# METAADAPTER: LEVERAGING META-LEARNING FOR EXPANDABLE REPRESENTATION IN FEW-SHOT CLASS INCREMENTAL LEARNING

## ABSTRACT

Few-shot class incremental learning (FSCIL) aims to enable models to learn new tasks from few labeled samples while retaining knowledge of previously ones. This scenario typically involves an offline base session with sufficient data for pre-training, followed by online incremental sessions where new classes are learned from limited samples. Existing methods either rely on a frozen feature extractor or meta-testing simulation to address overfitting issues in online sessions. However, they primarily learn feature representations using only the base session data, which significantly compromises the model's plasticity in feature representations. To enhance plasticity and reduce overfitting, we propose the MetaAdapter framework, which makes use of meta-learning for expandable representation. During the base session, we expand the network with pre-trained weights by inserting parallel adapters and employ meta-learning to encode generalizable knowledge into these modules. Then, the backbone is further trained on abundant data from the base classes to acquire fundamental classification ability. In each online session, the adapters are first initialized with parameters from meta-training, and subsequently tuned to adapt to the new classes. Leveraging meta-learning to produce initial adapters, MetaAdapter enables the feature extractor to effectively adapt to few-shot new classes, thus improving the generalization of the model. Experimental results on the mini-ImageNet, CUB200, and CIFAR100 datasets demonstrate that our proposed framework achieves the state-of-the-art performance.

## 1 INTRODUCTION

Deep neural networks have excelled in various vision tasks (Ren et al., 2015; He et al., 2016; Huang et al., 2017), but they usually depend on pre-collected datasets to achieve this success. However, data often arrives as a stream with continuously emerging new classes in real-world applications. An ideal model should recognize new categories while retaining the ability to distinguish previously learned ones, a process known as class-incremental learning (CIL) (Li & Hoiem, 2017; Rebuffi et al., 2017; Schwarz et al., 2018; Wu et al., 2019; Zhu et al., 2021a). The primary challenge in CIL is catastrophic forgetting (Goodfellow et al., 2015), where updating for new classes leads to forgetting old ones. The trade-off between maintaining performance on old categories (stability) and adapting to new ones (plasticity) is known as the stability-plasticity dilemma (Mermillod et al., 2013).

Traditional CIL methods typically assume abundant data for each new category, but this is often impractical in real-world applications due to the high costs of data collection and labeling. This challenge has driven the development of few-shot class incremental learning (FSCIL) (Tao et al., 2020). In FSCIL, the model is first pretrained on abundant data during the base session, while it must continually learn new classes from limited data in each incremental session. Similar to CIL, FSCIL also suffers from the stability-plasticity dilemma. Moreover, the limited availability of new class instances often results in overfitting (Zou et al., 2022), which means the model performs well on the training data in incremental sessions but has poor generalization performance on unseen data, thus reducing the model's generalization capability.

In the realm of few-shot learning (FSL), meta-learning enhances the learning effect of the current task by utilizing data from other related tasks (Rusu et al., 2019; Liu et al., 2020; Hospedales et al.,

2021). Building on these advancements in FSL, many approaches in FSCIL use meta-learning to reduce dependence on new data, which helps alleviate the risk of overfitting (Tian et al., 2024). These FSCIL approaches that rely on meta-learning can generally be divided into two categories: prototype-based and process-based methods. Prototype-based methods freeze the feature extractor following the base session and subsequently use it to generate prototypes for new classes, which either serve as classifier weights or align with the prototypes of base classes (Wang et al., 2024). Process-based methods typically mimic the meta-testing scenario by sampling a sequence of incremental tasks from base classes (Chi et al., 2022; Zhou et al., 2022b). Nonetheless, these two types of methods rely primarily on base session data to learn feature representations, which restricts the model's plasticity when adapting to new data (Zhang et al., 2023).

To reduce overfitting and improve model's plasticity, we introduce the MetaAdapter framework, a novel approach that integrates meta-initialized adapters to expand and enhance feature representations. Inspired by residual adapters (Rebuffi et al., 2018) used in domain adaptation (Zheng et al., 2021), we encode the task-agnostic knowledge into lightweight adapters, which are embedded as extensions of the backbone. The training process is divided into three phases, where the first two are conducted during the base session and the final phase focuses on few-shot adaptation in incremental sessions. During the first phase, we construct few-shot tasks by randomly sampling instances from each base class and train the adapters by meta-learning algorithms, such as Reptile (Nichol et al., 2018), to obtain generalizable initial parameters. During the second stage of backbone training, we introduce the feature compactness loss (FCL) to bring feature representations closer together, which prevents excessive dispersion in the embedding space, and thus reservs space for representation expansion. Additionaly, we search for flat local minima by adding gradient-based perturbations to the parameters to enhance the model's robustness against forgetting. For each incremental session in the third phase, the backbone is kept frozen and serves as the teacher model for knowledge distillation, while the adapters, initialized with parameters from the first phase, are tuned to expand the current representations to encompass new class features. With the meta-initialized adapters, MetaAdapter enables the model to adapt to new few-shot tasks efficiently without significantly increasing the architectural complexity. During the test stage, the backbone and adapters are fused through structural re-parameterization, ensuring that the model structure remains consistent during testing.

The contributions of this paper can be summarized as follows:

- We introduce a novel MetaAdapter framework, which incorporates meta-initialized adapters to expand and refine feature representations with the goal of effectively mitigating overfitting and improving the model's plasticity.
- A unique loss for FSCIL, called feature compactness loss, is proposed to prevent the feature space from becoming overly dispersed and leave more room for representation expansion.
- Extensive experiments on standard benchmarks CIFAR100, mini-ImageNet, and CUB200 show that our method outperforms baselines and achieves state-of-the-art results. Furthermore, we perform a thorough analysis to evaluate the importance of each component.

## 2 RELATED WORK

**Meta-learning.** Meta-learning, often described as learning how to learn, involves extracting insights from multiple learning episodes and using this knowledge to improve learning efficiency in future tasks (Hospedales et al., 2021). It is usually divided into two stages. During the meta-training stage, the model is trained using multiple source tasks to obtain initial network parameters with strong generalization ability. In the meta-testing stage, the model uses the parameters learned during meta-training to quickly adapt unseen tasks with only a few samples. Due to its natural suitability for FSL, meta-learning has been widely adopted in many studies (Triantafillou et al., 2018; Jamal & Qi, 2019; Elsken et al., 2020). In our study, we employ Reptile (Nichol et al., 2018), one of the most popular meta-learning algorithms, for adapter initialization to mitigate overfitting.

**Balancing Stability and Plasticity in Continual Learning.** In continual learning, a core challenge is the stability-plasticity dilemma, which involves balancing the model's consistent performance on learned classes (stability) and its adaptability to new classes (plasticity). Architecture-based methods have been widely explored to enhance plasticity by allowing automatic adjustment of network architecture during runtime. A popular choice is to separate network components into task-sharing and task-specific components, with the latter often being expandable. These task-specific components

often include parallel branches, such as ACL (Ebrahimi et al., 2020) and ReduNet (Wu et al., 2021); adaptive layers, including GVCL (Loo et al., 2020) and DyTox (Douillard et al., 2022); and low-rank factorization techniques, like RCM (Kanakis et al., 2020) and IBP-WF (Mehta et al., 2021). Another direction involves leveraging parallel sub-networks or sub-modules to learn incremental tasks without explicitly defining task-sharing or task-specific components. For instance, Progressive Neural Networks (Rusu et al., 2016) add identical sub-networks for each task, facilitating task-specific learning while allowing knowledge transfer through adaptor connections. And methods like PathNet (Fernando et al., 2017) and RPSNet (Jathushan et al., 2019) pre-allocate multiple parallel networks to construct a few candidate paths and select the best path for each task. However, dynamic neural networks often suffer from increased architectural complexity, which compromises their efficiency. In our study, we propose using lightweight, meta-initialized adapters, which allow the model to efficiently adapt to new few-shot tasks without significantly increasing the model's complexity.

**Few-Shot Class Incremental Learning (FSCIL).** As a variant of class-incremental learning (CIL), FSCIL requires rapid adaptation to new classes with limited data in each incremental session (Tao et al., 2020). Many FSCIL approaches build on advancements in FSL, particularly utilizing meta-learning to improve learning performance by leveraging data from related tasks (Rusu et al., 2019; Hospedales et al., 2021). The methods in FSCIL leveraging meta-learning can be broadly categorized into two types: prototype-based and process-based approaches. Prototype-based methods typically freeze the feature extractor trained on base classes and use the prototypes of new classes as the corresponding classifier weights (Zhang et al., 2021; Zhu et al., 2021b). While the frozen feature extractor helps alleviate overfitting problem, it often results in biased prototypes (Liu et al., 2020). Existing prototype adjustment methods (Liu et al., 2020; Zhu et al., 2021b; Zhang et al., 2021; Zhou et al., 2022a) aim to correct this bias but often involve complex pre-training algorithms (e.g., contrastive learning, data mixup) (Zhou et al., 2022a; Peng et al., 2022; Song et al., 2023). Process-based methods focus on meta-testing simulation by sampling sequences of incremental tasks from base classes (Yoon et al., 2020; Chi et al., 2022; Zhou et al., 2022b). For example, MetaFSCIL (Chi et al., 2022) adopts a meta-objective during the base phase to mimic the evaluation protocol through sequential task sampling. In contrast, our approach leverages meta-learning to produce meta-initialized adapters, offering a generalizable starting point for enhancing and refining feature representations. During the online incremental learning stage, MetaFSCIL uses Bi-directional Guided Modulation (BGM) to generate activation masks to mitigate forgetting. In comparison, our MetaAdapter framework keeps the backbone frozen and utilize it as a teacher model for knowledge distillation to guide the adaptation of lightweight adapters.

## 3 METHODOLOGY

We begin with the necessary problem setting in Section 3.1, followed by an overview of the framework in Section 3.2. Sections 3.3 to 3.5 cover the specific training process.

### 3.1 PROBLEM SETTING

The aim of FSCIL is to accommodate new knowledge from limited samples of novel classes and resist forgetting previously learned old classes. We assume there exists $T$ sessions in total, including a base session (*i.e.*, the first session) and $T-1$ incremental sessions (*i.e.*, sessions after the first session). The training data in the base session is denoted as $\mathcal{D}^0$, and the training data in the incremental sessions is represented as $\{\mathcal{D}^1, \mathcal{D}^2, \ldots, \mathcal{D}^{T-1}\}$. For the training data $\mathcal{D}^t$ in the $t$-th session, it is denoted by $\{(x_i, y_i)\}_{i=1}^{N_t}$ with the corresponding label space $\mathcal{C}^t$. Note that the training label space between different sessions are disjoint, *i.e.*, for any $i, j \in \{0, 1, \ldots, T-1\}$ and $i \neq j, \mathcal{C}^i \cap \mathcal{C}^j = \varnothing$. Following standard incremental learning paradigm, a model in each session $t$ can only access $\mathcal{D}^t$. Usually, the training set $\mathcal{D}^0$ in the base session contains a sufficient volume of data for base classes in $\mathcal{C}^0$. In contrast, each training set $\mathcal{D}^t(1 \leq t \leq T-1)$ in the following sessions contains few training samples, which can be denoted as a $N$-way $K$-shot classification task, comprising of only $K$ examples for each of the $N$ categories from $\mathcal{C}^t$. Once the incremental learning in session $t$ is finalized, the model is tested on query samples from all the seen classes so far: $\tilde{\mathcal{C}}^t = \mathcal{C}^0 \cup \mathcal{C}^1 \cdots \cup \mathcal{C}^t$.

In this work, the model is decoupled into a feature encoder $\phi_\theta(\cdot)$ with parameters $\theta$, and a linear classifier $W$. Given a sample $x_j \in \mathbb{R}^D$, the feature representation of $x_j$ is denoted as

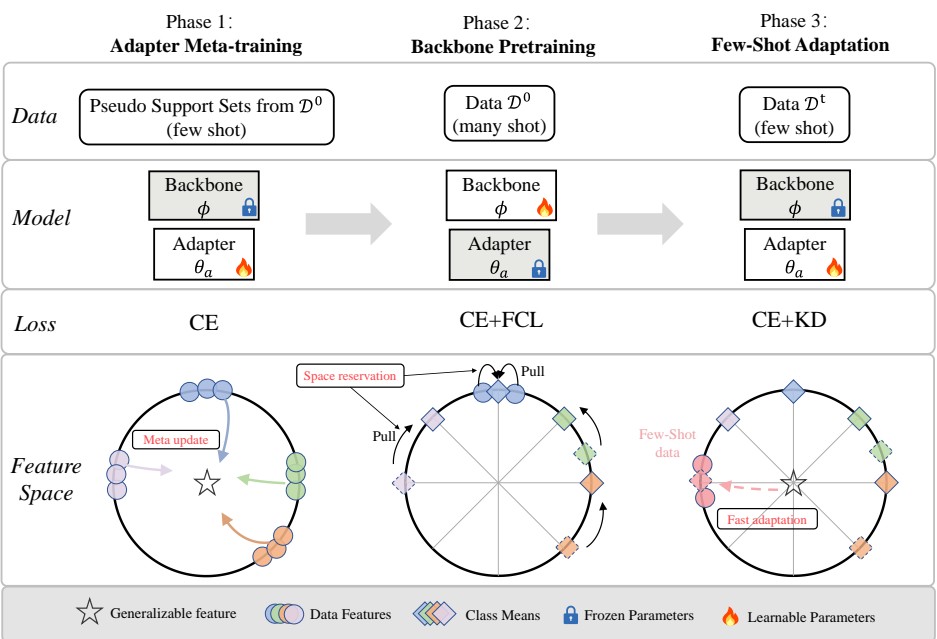

Figure 1: Overview of MetaAdapter framework. Our MetaAdapter framework is a three-phase approach for FSCIL. Phase 1: (Adapter Meta-training) Few-shot tasks are constructed by sampling instances from each base class, and the adapters are trained using meta-learning. Phase 2: (Backbone Pretraining) To reserve space for future tasks, we introduce FCL to keep the embedding space compact during backbone training. Phase 3: (Few-Shot Adaptation) We only fine-tunes these adapters to learn the new task while keeping the pre-trained weight of the backbone.

$\phi_\theta(x_j) \in \mathbb{R}^d$. For an $N$-class classification task, the output logits of the sample $x_j$ are given by $\mathcal{O}_j = W^\top \phi_\theta(x_j) \in \mathbb{R}^N$, where $W \in \mathbb{R}^{d \times N}$.

### 3.2 OVERVIEW OF METAADAPTER FRAMEWORK

Our MetaAdapter framework for FSCIL (see Figure 1) begins with a meta-training of the adapters in the first phase, with the aim of obtaining generalizable initial parameters for future few-shot tasks. In the subsequent second phase, referred to as backbone pretraining in Figure 1, we leverage feature compactness loss (FCL) to enhance the similarity of sample feature representations using abundant data from $\mathcal{D}^0$, which equips the model with fundamental classification ability and prevents the feature space from becoming overly dispersed. The third phase, implemented in each subsequent incremental session, uses meta-initialized adapters to rapidly adapt to new few-shot tasks while preserving old knowledge retained in the backbone.

### 3.3 ADAPTER META-TRAINING

In FSCIL, a model needs to adapt to new classes with limited instances and then evaluate over all seen classes. As a result, a more generalizable feature will facilitate more effective learning of future classes and improve overall performance. This indicates that during the offline training phase, the model needs to be trained on multiple source tasks to acquire initial parameters capable of strong generalization, which enables effective adaptation to new categories in incremental sessions. To this end, we construct pseudo support sets by randomly sampling instances from each base class and apply meta-learning on these sets to search for an effective initialization.

Specifically, we expand the model's structure by modularly adding adapters to learn new classes and then apply the Reptile algorithm to these modules with the support sets. To make the pseudo few-shot support sets share the similar data format as online incremental tasks, we partition the base label space $Y_0$ into non-overlapping sets: $Y_0 = \hat{Y}_1 \cup \hat{Y}_2 \cup \cdots \cup \hat{Y}_C$, where $|\hat{Y}_i| = \hat{N}$ and $|Y_0| = \hat{N}C$. We then randomly sample $\hat{K}$ examples from the corresponding label space $\hat{Y}_i$ to construct an $\hat{N}$-way

$\hat{K}$-shot training set $\mathcal{S}^i$, forming a sequence of support sets $\mathcal{S}^1, \mathcal{S}^2, \ldots, \mathcal{S}^C$. The adapter parameters $\theta_a$ are first initialized with a shared random initialization across all pseudo tasks. Then, we start to perform fast adaptation to new classes and obtain $\theta_a^j$ via a few gradient steps:

$$\theta_a^j \leftarrow \theta_a - \alpha \nabla_\theta \mathcal{L}_{\text{CE}}(\mathcal{X}_j^s, \mathcal{Y}_j^s; \theta_a), \tag{1}$$

where $\mathcal{X}_j^s$ and $\mathcal{Y}_j^s$ are the samples and labels for support set in the $j$-th task, $\alpha$ is the learning rate for task-specific updates, and $\mathcal{L}_{\text{CE}}$ denotes the cross-entropy loss. After completing the gradient descent updates for all tasks, the parameter initialization in the meta-learner is updated as follows:

$$\theta_a \leftarrow \theta_a + \beta \cdot \frac{1}{C} \sum_{j=1}^{C} (\theta_a^j - \theta_a), \tag{2}$$

where $\beta$ is the learning rate for meta-updates, and $C$ is the total number of tasks.

### 3.4 BACKBONE PRETRAINING

**Feature Compactness Loss.** Traditional pre-training methods often lead to a dispersed embedding space, as they focus on optimizing empirical loss and maximizing inter-class margins for base-class prototypes. While these strategies enhance feature discrimination, they may result in overfitting to base classes and reduced adaptability to few-shot new classes. To mitigate these issues and reserve capacity for incremental learning, we propose the Feature Compactness Loss (FCL). By compacting both inter-class and intra-class distances, FCL prevents the embedding space from becoming overly dispersed, which preserves learning capacity for future few-shot incremental learning scenarios.

We employ the FCL on the set $O$ predicated on the current batch. $O$ contains the following: mean features for all classes within batch $\mathbf{c}_{\text{batch}}$, mean features of unseen classes within batch $\mathbf{c}_{\text{unseen}}$ and original features whith batch $\mathbf{p}$. For every training batch $B$ and their corresponding class labels $y$, the within-batch means for all data features are computed as:

$$\mathbf{c}_{\text{batch},k} = \frac{1}{\text{Num}_k} \sum_{y_j=k} \mathbf{p}_j, \tag{3}$$

where $\mathbf{c}_{\text{batch},k}$ represents the mean feature of category $k$ in the batch, and $\text{Num}_k$ is the number of samples in category $k$. Mean features of unseen classes in the current batch are derived from the average feature representations computed during the previous epoch. The set $O$ is defined as $O = \{\mathbf{c}_{\text{batch}} \cup \mathbf{c}_{\text{unseen}} \cup \mathbf{p}\}$. Then we compute pairwise cosine similarities between all feature vectors in $\mathbf{p}_{\text{concat}}$, and the Feature Compactness Loss (FCL) takes the form:

$$\mathcal{L}_{\text{fcl}} = -\frac{1}{|O|} \sum_{i=1}^{|O|} \sum_{j \neq i} \log \sigma \left( \frac{\text{sim}(\mathbf{v}_i, \mathbf{v}_j)}{\tau} \right), \tag{4}$$

where $\mathbf{v}_i$ and $\mathbf{v}_j$ are feature vectors from the composite feature set $O$, $\text{sim}(\mathbf{x}, \mathbf{y}) = \frac{\mathbf{x}^\top \mathbf{y}}{\|\mathbf{x}\|\|\mathbf{y}\|}$ denotes the cosine similarity between two vectors, $\sigma(\cdot)$ is the softmax function, and $\tau$ is a temperature parameter that controls the smoothness of the probability distribution. This approach leaves more space within the current feature representations for expansion, which facilitates the accommodation of new categories during the incremental learning phase.

Finally, the total loss function in the base session is the weighted sum of the FCL and standard cross-entropy loss:

$$\mathcal{L}_{\text{base}} = w_{\text{fcl}} \cdot \mathcal{L}_{\text{fcl}} + \mathcal{L}_{\text{CE}}, \tag{5}$$

where $w_{\text{fcl}}$ is a hyperparameter that balances the contributions of the two losses.

**Searching for Flat Local Minima.** To mitigate the interference of adapter integration on parameters and enhance the model's resilience to forgetting, we employ the SAM method (Foret et al., 2021) during the base training phase. SAM refines the base loss function by searching for flat minima, which reduces the model's sensitivity to small perturbations in the data and consequently enhancing

Figure 2: (a) Illustration of our proposed method in the incremental stage, where modules with dashed lines are used only in the forward pass with frozen parameters. (b) Visualization of expanded classification weights in session $t$.

overall performance. Specifically, SAM operates by exploring a $\rho$-ball neighborhood during each parameter update. It first identifies a small perturbation that maximizes the base loss:

$$\epsilon(\theta) = \rho \frac{\nabla \mathcal{L}_{\text{base}}(\theta)}{\|\nabla \mathcal{L}_{\text{base}}(\theta)\|}, \tag{6}$$

$$\mathcal{L}_{\text{base}}^{\rho}(\theta) = \mathcal{L}_{\text{base}}(\theta + \epsilon(\theta)). \tag{7}$$

Subsequently, gradient descent is applied to the perturbed parameters:

$$\nabla \mathcal{L}_{\text{base}}^{\rho}(\theta) = \nabla \mathcal{L}_{\text{base}}(\theta + \epsilon(\theta)). \tag{8}$$

This process helps the model converge to a flatter solution, leading to greater stability and better generalization during incremental learning.

### 3.5 FEW-SHOT ADAPTATION

**Main Branch Distillation.** As shown in Figure 2(a), before each incremental learning phase, we first perform structure expansion by expanding adapters and initialize the new parameters using those obtained from meta-training in the base session. In each session $t$ of the incremental phase, to further handle new classes, the classification weights $W_n^{t-1}$ from session $t-1$ are expanded to $\hat{W}_n^{t-1}$ (shape $\mathbb{R}^{d \times |\tilde{\mathcal{C}}^{t-1}|} \to \mathbb{R}^{d \times |\tilde{\mathcal{C}}^t|}$) based on $\mathcal{D}^t$ as shown in Figure 2(b). Specifically, the classification weights for the newly appeared classes in session $t$ are computed using the feature centroids of training samples with the same labels:

$$\boldsymbol{w}_c = \frac{1}{N_c} \sum_{(\mathbf{x}_i, y_i) \in \mathcal{D}^t} \mathbb{I}\left[y_i = c\right] \phi_\theta\left(\mathbf{x}_i\right), \tag{9}$$

where $\boldsymbol{w}_c$ is the prototype of class $c$, $\mathbb{I}$ is the indicator function, and $N_c$ is the number of samples in class $c$ in $\mathcal{D}^t$. The adapter and classification weights of the incremental branch are then initialized as $\{\theta_a^{t-1}, \hat{W}_n^{t-1}\}$, and these parameters can be further fine-tuned to accommodate new classes.

By utilizing the residual structure, the adapters can retain the generalization capabilities from the previous model while adapting to new tasks. However, the decision boundary often shifts towards the new classes, which can result in poor performance on previous classes. To ensure the updated model can still classify instances of the old classes, we refer to the backbone and apply knowledge distillation loss to implicitly constrain parameter updates. The loss for distillation is defined as:

$$\mathcal{L}_{\text{kd}} = -\sum_{k=1}^{|\tilde{\mathcal{C}}^t|} \tau_k(\mathbf{z}_n^{t-1}) \log \tau_k(\mathbf{z}_n^t), \tag{10}$$

where $\mathbf{z}_n^{t-1}$ and $\mathbf{z}_n^t$ represent the logits from the previous and current models, respectively. The function $\tau_k(\mathbf{z}) = \frac{e^{\gamma \cdot \mathbf{z}(k)}}{\sum_{j=1}^{|\tilde{\mathcal{C}}^t|} e^{\gamma \cdot \mathbf{z}(j)}}$ denotes a temperature-scaled softmax output, with $\mathbf{z}(k)$ as the

Table 1: Performance of FSCIL in each session on mini-ImageNet and comparison with other methods. "Avg." is the average accuracy of all sessions. "PD" denotes the performance drop, i.e., the accuracy difference between the first and the last sessions. "Final Improv." calculates the improvement of our method in the last session.

| Backbone | Methods | Accuracy in each session (%) | | | | | | | | | Avg. | PD | Final Improv. |
|---|---|---|---|---|---|---|---|---|---|---|---|---|---|
| | | 0 | 1 | 2 | 3 | 4 | 5 | 6 | 7 | 8 | | | |
| ResNet-18 | iCaRL (Rebuffi et al., 2017) | 61.31 | 46.32 | 42.94 | 37.63 | 30.49 | 24.00 | 20.89 | 18.80 | 17.21 | 33.29 | 44.10 | **+42.04** |
| | TOPIC (Tao et al., 2020) | 61.31 | 50.09 | 45.17 | 41.16 | 37.48 | 35.52 | 32.19 | 29.46 | 24.42 | 39.64 | 36.89 | **+34.83** |
| | ERL++ (Dong et al., 2021) | 61.70 | 57.58 | 54.66 | 51.72 | 48.66 | 46.27 | 44.67 | 42.81 | 40.79 | 49.87 | **20.91** | **+18.46** |
| | CEC (Zhang et al., 2021) | 72.00 | 66.83 | 62.97 | 59.43 | 56.70 | 53.73 | 51.19 | 49.24 | 47.63 | 57.75 | 24.37 | **+11.62** |
| | F2M (Shi et al., 2021) | 72.05 | 67.47 | 63.16 | 59.70 | 56.71 | 53.77 | 51.11 | 49.21 | 47.84 | 57.89 | 24.21 | **+11.41** |
| | Replay (Liu et al., 2022) | 71.84 | 67.12 | 63.21 | 59.77 | 57.01 | 53.95 | 51.55 | 49.52 | 48.21 | 58.02 | 23.63 | **+11.04** |
| | MetaFSCIL (Chi et al., 2022) | 72.04 | 67.94 | 63.77 | 60.29 | 57.58 | 55.16 | 52.90 | 50.79 | 49.19 | 58.85 | 22.85 | **+10.06** |
| | FACT (Zhou et al., 2022a) | 75.32 | 70.34 | 65.84 | 62.05 | 58.68 | 55.35 | 52.42 | 50.42 | 48.51 | 59.88 | 26.81 | **+10.74** |
| | TEEN (Wang et al., 2024) | 74.85 | 70.65 | 66.50 | 62.88 | 60.38 | 57.34 | 54.71 | 53.06 | 51.70 | 61.34 | 23.15 | **+7.55** |
| | ALICE (Peng et al., 2022) | 80.60 | 70.60 | 67.40 | 64.50 | 62.50 | 60.00 | 57.80 | 56.80 | 55.70 | 63.99 | 24.90 | **+3.55** |
| | BiDist (Zhao et al., 2023) | 74.65 | 69.89 | 65.44 | 61.76 | 59.49 | 56.11 | 53.28 | 51.74 | 50.49 | 60.32 | 24.16 | **+8.76** |
| | CEC+ (Wang et al., 2023) | 82.65 | 77.82 | 73.59 | 70.24 | 67.74 | 64.82 | 61.91 | 59.96 | 58.35 | 68.56 | 24.30 | **+0.90** |
| | MetaAdapter (ours) | **82.80** | **78.46** | **74.39** | **71.57** | **68.71** | **65.69** | **63.40** | **60.63** | **59.25** | **69.43** | 23.55 | |
| ResNet-12 | NC-FSCIL (Yang et al., 2023) | 84.02 | 76.80 | 72.00 | 67.83 | 66.35 | 64.04 | 61.46 | 59.54 | 58.31 | 67.82 | 25.71 | **+2.70** |
| | C-FSCIL (Hersche et al., 2022) | 76.40 | 71.14 | 66.46 | 63.29 | 60.42 | 57.46 | 54.78 | 53.11 | 51.41 | 61.61 | 25.00 | **+9.60** |
| | OrCo (Ahmed et al., 2024) | 83.30 | 70.80 | 66.90 | 64.32 | 62.28 | 60.46 | 58.40 | 58.02 | 58.08 | 64.73 | 25.22 | **+2.93** |
| | MetaAdapter (ours) | **84.12** | **79.95** | **75.97** | **72.61** | **69.68** | **66.88** | **64.12** | **62.39** | **61.01** | **70.75** | **23.11** | |

$k$-th element of $\mathbf{z}$ and $\gamma$ as the temperature coefficient controlling the sharpness of the distribution. We formulate the final loss function for adapter tuning as:

$$\mathcal{L}_{\text{incre}} = \mathcal{L}_{\text{CE}} + w_{\text{kd}} \cdot \mathcal{L}_{\text{kd}}, \tag{11}$$

where $w_{\text{kd}}$ is a trade-off hyperparameter.

**Adapter Integration.** Following the training in each incremental learning phase, the parameters of the adapters are integrated into the corresponding convolutional layers to maintain the model's architecture. This process not only enhances the model's representational capacity but also ensures that the number of network parameters remains constant across phases. Specifically, by zero-padding and linear transformation, the parameters in the residual structure are fused with the original convolution kernel parameters. Before leanring the $t$-th new task ($t > 1$), MetaAdapter maintains the weight $\theta_{\text{conv}}^t$. After learning the $t$-th task, MetaAdapter integrates the $t$-th branch into $\theta_{\text{conv}}^t$ and obtains:

$$\theta_{\text{conv}}^t = \theta_{\text{conv}}^{t-1} + w_a \cdot F_{\text{pad}}(\theta_a^t), \tag{12}$$

where $\theta_{\text{conv}}^{t-1}$ represents the convolutional layer parameters from the previous phase, $\theta_a^t$ represents the adapter layer parameters at phase $t$, $w_a$ denotes the adapter weight, and $F_{\text{pad}}$ is a padding function used to match the dimensions of the adapter weights with the convolution weights of the backbone. In this way, the adapter parameters in previous tasks do not need to be maintained in the learning of subsequent tasks. Therefore, throughout the learning process, MetaAdapter only expands the number of parameters by adding a single branch of adapters alongside the backbone during training, while for testing, the number of parameters remains the same as the backbone alone. The final integrated parameters are represented as:

$$\theta_{\text{conv}}^{\text{final}} = \theta_{\text{conv}}^{\text{base}} + w_a \cdot \sum_{t=1}^{T-1} F_{\text{pad}}(\theta_a^t). \tag{13}$$

Details regarding the impact of different adapter structures on model performance and the specific configuration used in our experiments can be found in Appendix B.

## 4 EXPERIMENTS

In this section, we evaluate our method on FSCIL benchmark datasets, including mini-ImageNet (Russakovsky et al., 2015), CIFAR100 (Krizhevsky et al., 2009), and CUB200 (Wah et al., 2011),

Table 2: Ablation studies on three datasets to investigate the effects of our proposed method. "FINAL" refers to the accuracy of the last session; "AVERAGE" is the average accuracy of all sessions; "PD" denotes the performance drop, i.e., the accuracy difference between the first and the last sessions.

| MIS | FCL | SAM | mini-ImageNet | | | CIFAR100 | | | CUB200 | | |
|---|---|---|---|---|---|---|---|---|---|---|---|
| | | | FINAL↑ | AVERAGE↑ | PD↓ | FINAL↑ | AVERAGE↑ | PD↓ | FINAL↑ | AVERAGE↑ | PD↓ |
| ✓ | | | 56.96 | 66.34 | 25.11 | 57.45 | 68.05 | 25.97 | 60.69 | 67.61 | 19.31 |
| | ✓ | | 60.67 | 70.05 | 23.29 | 58.75 | 69.15 | 25.35 | 60.35 | 67.69 | 20.23 |
| | ✓ | ✓ | 60.66 | 70.44 | 23.70 | 58.08 | 69.22 | 25.77 | 60.66 | 67.75 | 19.92 |
| ✓ | | ✓ | 56.82 | 66.40 | 25.61 | 56.92 | 67.48 | 26.38 | 60.82 | 67.53 | 19.21 |
| ✓ | ✓ | | 60.36 | 70.17 | 23.99 | 58.84 | 69.11 | 25.11 | 60.92 | 67.89 | 19.52 |
| ✓ | ✓ | ✓ | **61.01** | **70.75** | **23.11** | **59.20** | **69.73** | **24.85** | **61.70** | **68.45** | **18.93** |

and compare it with state-of-the-art methods. We also perform ablation studies to validate each component. Detailed experimental setups can be found in Appendix A.

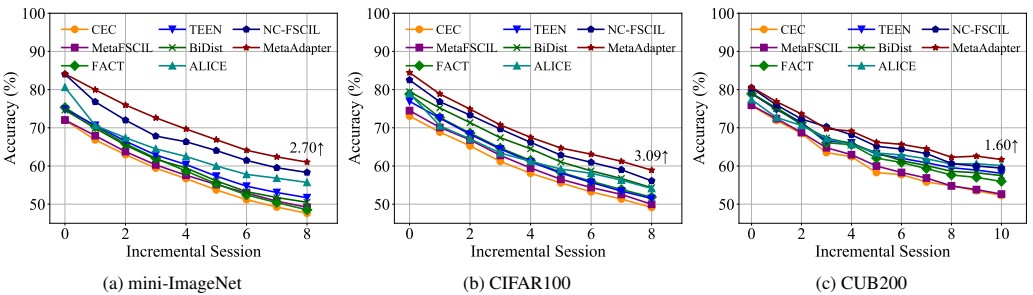

(a) mini-ImageNet      (b) CIFAR100      (c) CUB200

Figure 3: Performance curves of our method comparing to state of-the-art FSCIL methods on three datasets. We annotate the performance gap after the last session between MetaAdapter and the runner-up method at the end of each curve.

### 4.1 PERFORMANCE ON BENCHMARKS

We present the accuracy after each session for the benchmark datasets mini-ImageNet, CIFAR100, and CUB200 in Figure 3, with detailed experimental results provided in Table 1, Table 5 (Appendix B), and Table 6 (Appendix B), respectively. As shown in Figure 3, our method consistently achieves superior performance across all sessions on mini-ImageNet, CIFAR100, and CUB200 compared to previous studies. Specifically, compared to the NC-FSCIL method (Yang et al., 2023), which has demonstrated strong performance in FSCIL, our approach improves the average performance by 2.93% on mini-ImageNet, 1.87% on CIFAR100, and 1.17% on CUB200. We also achieve a final accuracy increase of over 2.7% on both mini-ImageNet and CIFAR100, and an improvement of 2.26% on CUB200. The above observations indicate that our method can effectively adapt to novel classes with limited data and enhance generalization ability of the model.

### 4.2 ABLATION STUDIES

We first validate our three key components: the meta-initialization strategy (MIS) for adapters in Section 3.3, the feature compactness loss (FCL) and sharpness-aware minimization (SAM) in Section 3.4. Following this, we conduct hyper-parameter sensitivity test experiments.

**Validation of Key Components.** We conducted experiments with different combinations of our three key components to evaluate their individual and collective contributions to the model's performance. From the results in Table 2, we can observe the contribution of each component to the overall performance. When only MIS is applied, the model demonstrates significant improvements compared to other methods, particularly in maintaining a high average accuracy across all datasets. However, the addition of FCL and SAM further enhances the performance by reducing the performance drop (PD) between sessions. Across mini-ImageNet, CIFAR100, and CUB200, adding both FCL and SAM consistently yields the lowest PD values of 23.11, 24.85, and 18.93, respectively, while also achieving the highest final accuracies of 61.01, 59.20, and 61.70. These results confirm

Table 3: Base, Incremental, and Harmonic Mean accuracy across sessions on CIFAR100. "Avg." is the average accuracy of all sessions. The Harmonic Mean, following the work of Peng et al. (2022), is used to evaluate the balanced performance between the base and new classes.

| Method | Class Group | Accuracy in each session (%) | | | | | | | | | Avg. |
|---|---|---|---|---|---|---|---|---|---|---|---|
| | | 0 | 1 | 2 | 3 | 4 | 5 | 6 | 7 | 8 | |
| NC-FSCIL (Yang et al., 2023) | Base | 82.52 | 79.55 | 78.63 | 77.98 | 77.60 | 75.98 | 74.45 | 75.138 | 73.98 | 77.32 |
| | Incremental | - | 44.00 | 41.60 | 36.47 | 31.95 | 31.32 | 33.97 | 31.31 | 29.30 | 34.99 |
| | Harmonic Mean | - | 56.66 | 54.41 | 49.70 | 45.26 | 44.36 | 46.65 | 44.21 | 41.98 | 47.90 |
| MetaAdapter (ours) | Base | 84.05 | 81.73 | 80.50 | 79.53 | 78.48 | 77.42 | 76.78 | 76.23 | 75.17 | 78.88 |
| | Incremental | - | 44.40 | 43.10 | 40.07 | 37.70 | 36.24 | 37.07 | 36.29 | 35.25 | 38.76 |
| | Harmonic Mean | - | 57.54 | 56.14 | 53.29 | 50.93 | 49.37 | 49.99 | 49.17 | 47.99 | 51.80 |

that the combination of MIS, FCL, and SAM significantly improves the model's performance by balancing stability and plasticity across sessions.

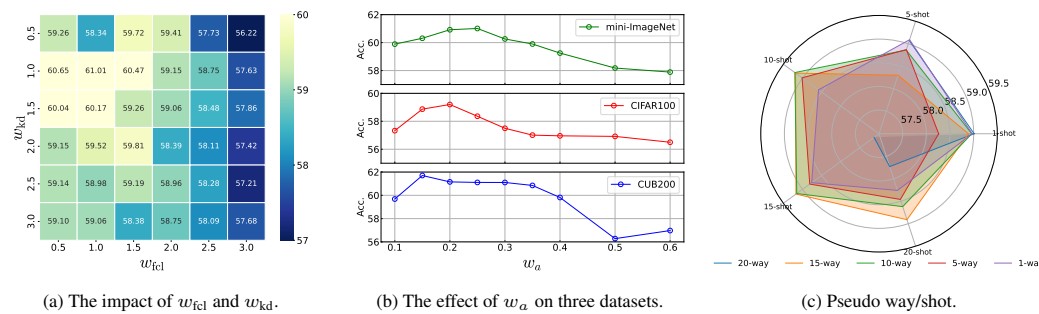

(a) The impact of $w_{\text{fcl}}$ and $w_{\text{kd}}$.      (b) The effect of $w_a$ on three datasets.      (c) Pseudo way/shot.

Figure 4: Hyper-parameter Analysis.

**Hyper-Parameter Sensitivity.** In Eq. (5), Eq. (11) and Eq. (12), three key hyper-parameters are involved during training: $w_{\text{fcl}}$, $w_{\text{kd}}$ and $w_a$. We conducted a hyper-parameter search to determine suitable values for $w_{\text{fcl}}$ and $w_{\text{kd}}$ by exploring their combinations. As observed from Figure 4(a), our model achieves satisfactory performance on the mini-ImageNet dataset when using a relatively larger value for $w_{\text{kd}}$ and a smaller value for $w_{\text{fcl}}$. Moreover, the model's performance remains robust across a wide range of values for these two hyperparameters, with $w_{\text{kd}}$ ranging from 1.0 to 3.0 and $w_{\text{fcl}}$ ranging from 0.5 to 2.0. This is because the model can prevent more knowledge from being forgotten with the help of a larger knowledge distillation weight $w_{\text{kd}}$. However, while the feature compactness loss can help to prevent feature representations from becoming overly dispersed, an excessively large coefficient for $w_{\text{fcl}}$ can negatively impact the classification performance. Additionally, consistent experimental results are observed across the other two datasets. As a result, we set $w_{\text{fcl}} = 1.0$ and $w_{\text{kd}} = 1.0$ respectively throughout our experiments.

In addition, we analyze the final accuracy fluctuation under different adapter weights $w_a$ which controls the strength of feature representation adjustments by the adapters in Eq. (12). The larger value of $w_a$ enhances the model's attention to the knowledge brought by the newly expanded features. As depicted in Figure 4 (b), the accuracy improves initially as $w_a$ increases, which allows the model to better learn new class knowledge. However, as $w_a$ becomes excessively large, the expanded feature representations start to interfere with the performance of the original backbone, leading to a decline in accuracy. The optimal $w_a$ for three datasets is 0.15, 0.20 and 0.25, respectively. For classes from the fine-grained dataset CUB200 that share similar appearance, a smaller $w_a$ is required to leverage more knowledge from the base stage. To the contrary, for classes from mini-ImageNet and CIFAR100, which are less semantically related, a relatively larger $w_a$ is preferable.

### 4.3 FURTHER ANALYSES

**Pseudo Way/Shot.** As discussed in Section 3.3, MetaAdapter requires sampling few-shot tasks from the base data for the meta-training stage. We vary the pseudo-incremental way across $\{1, 5, 10, 15, 20\}$ and the pseudo-incremental shot across the same values, resulting in 25 different configurations to evaluate their influence on the final accuracy on CIFAR100. We can infer

from Figure 4(c) that MetaAdapter with prefer moderate pseudo-training way and shot settings, *i.e.*, training with 10-way and 10-shot achieves best performance, with an accuracy of 59.20%. Furthermore, it is also evident that the influence of the pseudo-incremental way is stronger than that of the pseudo-incremental shot.

**Layer Locations for Adapter Placement.** We also conduct ablation experiments to explore the impact of selecting different residual layers for adapter insertion, focusing on the residual blocks of the ResNet architecture as described by He et al. (2016). As shown in Table 4, inserting adapters only into the last residual layer of conv5_x yields a final accuracy of 58.96%, highlighting the model's limited capacity to adapt to new tasks. On the other hand, placing adapters in all residual layers increases the number of parameters excessively, but slightly lowers accuracy to 58.81%, suggesting a risk of overfitting due to excessive parameters especially in few-shot scenarios. Finally, our model achieves the best trade-off between

Table 4: Impact of residual layer selection for adapter on CUB200.

| Layer Locations | Final Acc. |
|---|---|
| last resblock of conv5_x | 58.96 |
| conv5_x | 59.21 |
| conv4$\sim$5_x | 60.76 |
| conv3$\sim$5_x | 59.99 |
| conv3$\sim$4_x | **61.70** |
| conv2$\sim$4_x | 60.66 |
| conv2$\sim$5_x | 60.11 |
| all resblocks of backbone $\phi_\theta$ | 58.81 |

stability and plasticity when adapting the intermediate residual layers (*i.e.*, conv3_x to conv4_x). This is because the earlier layers are primarily involved in general feature extraction, while the later layers have a more significant impact on the classification performance of previous tasks.

**Trade-off between Base and Novel Classes.** For a deeper understanding of the challenges in FS-CIL, we analyze the model's ability to adapt to novel classes while preserving base knowledge by examining the individual accuracy of both base and novel classes, along with the harmonic mean. Table 3 shows that our approach outperforms the second-best result on novel classes by 6% in the final session which highlights the effectiveness of the meta-initialization strategy for adapters. At the same time, we still maintain competitive base class accuracy, as the adapter integration shows no significant forgetting. Finally, the highest harmonic mean demonstrates that our approach achieves a superior balance between performance on base and novel classes.

## 5 CONCLUSION

In this paper, we present a novel framework that leverages meta-initialized adapters to expand and strengthen feature representations for FSCIL. By applying meta-learning during the base session, we effectively train the adapters to capture generalizable knowledge, enabling the model to learn efficiently from limited task samples. Furthermore, a novel loss function is used to drive features closer together and prevent excessive dispersion in the embedding space. During incremental sessions, we tune the adapters to refine feature representations, which allows the model to effectively accommodate new knowledge. Through extensive experiments and comprehensive analysis, our approach consistently surpasses previous methods and sets a new state-of-the-art.

### STATEMENTS

**Ethics Statement.** This study does not involve any of the potential issues such as human subject, public health, privacy, fairness, security, *etc*. All authors affirm compliance with the ICLR Code of Ethics.

**Reproducibility Statement.** All datasets used in this paper are public and have been cited. Please refer to Appendix A for the dataset descriptions and the implementation details of our experiments. The source code necessary for reproducing all results is provided as part of the supplementary materials.

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

## A APPENDIX: EXPERIMENTAL SETUPS

**Datasets and Evaluation.** Following mainstream settings, our experiments are conducted on three benchmark datasets: mini-ImageNet (Russakovsky et al., 2015), CIFAR100 (Krizhevsky et al., 2009), and CUB200 (Wah et al., 2011). mini-ImageNet is a variant of ImageNet, including 60,000 images with an image size of $84 \times 84$ from 100 chosen classes. CIFAR100 is composed of 60,000 tiny images of size $32 \times 32$ from 100 categories. CUB200 is a fine-grained classification dataset for 200 bird species with similar appearance in a resolution of $224 \times 224$. For mini-ImageNet and CIFAR100, 60 categories are selected as base classes ($t = 0$) while the remaining are split into 8 incremental sessions ($1 \leq t \leq 8$) with only 5 training examples per novel class (i.e., 5-way 5-shot). As for the CUB200 dataset, 100 categories are selected as the base training sets, while the rest forms 10-way 5-shot tasks for 10 sessions in total.

**Training Details.** Previous studies commonly use ResNet-12, ResNet-18, and ResNet-20 (He et al., 2016) for FSCIL experiments. For mini-ImageNet and CIFAR100, we use ResNet-12 following (Hersche et al., 2022; Yang et al., 2023), and we first pre-train the model on half of the base classes to initialize the feature extractor. For CUB200, we use ResNet-18 (pre-trained on ImageNet) following other studies. In the base session, we first meta-train the adapters for 30 epochs across all datasets. We set $\alpha = 0.05, \beta = 0.01$ for CIFAR100, $\alpha = 0.01, \beta = 0.1$ for mini-ImageNet and $\alpha = 0.001, \beta = 0.001$ for CUB200. Following the meta-training of adapters, the backbone is trained on base session data. For CIFAR100 and mini-ImageNet, the training is conducted with a learning rate of 0.1, a batch size of 256, and for 1000 epochs. The cosine scheduler is used to adjust the learning rate. For CUB200, the backbone is trained with a learning rate of 0.004, a batch size of 128, and epochs of 400. In each incremental session, we further tune the adapters for 1-5 iterations using a learning rate of 0.001 on CUB200, 0.01 on mini-ImageNet, and 0.05 on CIFAR100. Before computing the cross-entropy loss, a commonly used temperature scalar is applied to adjust the distribution of the output logits. For example, the original output logits of instance $x_j$ are denoted as $\mathcal{O}_j \in \mathbb{R}^D$ . The logits used to compute cross-entropy loss are denoted as $\frac{\mathcal{O}_j}{\tau_o}$. The $\tau_o$ is set to 64 for mini-ImageNet and CIFAR100 datasets and 32 for CUB200 dataset. The selection of other hyper-parameters is provided in Section 4.2.

## B APPENDIX: MORE RESULTS

Our results on CIFAR100, as shown in Table 5, indicate that MetaAdapter outperforms all compared methods in terms of final accuracy improvement, achieving a notable gain of 69.73%. Similarly, our experimental results on CUB200, shown in Table 6, demonstrate that we achieve better accuracy across all sessions compared to most baseline methods. Although we do not surpass NC-FSCIL in the third session, we still maintain the highest average accuracy among all methods.

**Impact of Adapter Structure.** To investigate how the structure of residual adapters affects expandable representation during training, we design the following experiments. We evaluate four different convolutional block structures within the residual component: $1 \times 1$ convolution, a combination of $1 \times 1$ convolution with BatchNorm, $3 \times 3$ convolution, and a combination of $3 \times 3$ convolution with BatchNorm. As shown in Table 7, the performance of the $1 \times 1$ convolution and the combination are similar, while the $3 \times 3$ convolution results in slightly lower accuracy. This indicates that the $1 \times 1$ convolu-

Table 7: Performance under different expanding structures on CUB200.

| Adapter Structure | FINAL↑ | AVERAGE↑ | PD↓ |
|---|---|---|---|
| $3 \times 3$ conv | 59.11 | 67.37 | 21.33 |
| $3 \times 3$ conv + bn | 60.43 | 67.84 | 20.02 |
| $1 \times 1$ conv | **61.70** | **68.45** | **18.93** |
| $1 \times 1$ conv + bn | 61.52 | 68.32 | 19.28 |

tion structure is sufficient for learning the representation of new classes without requiring additional parameters. To ensure consistency in feature map dimensions during integration, the stride and padding configurations of the $1 \times 1$ adapters are aligned with the backbone convolutional layers. For example, if the backbone uses a $3 \times 3$ convolutional kernel with padding = 1, the $1 \times 1$ adapter kernel is aligned at the center with no additional padding (padding = 0). This ensures that the integrated kernel produces the same output feature maps as the original backbone.

Table 5: Performance of FSCIL in each session on CIFAR100 and comparison with other methods. "Avg." is the average accuracy of all sessions. "PD" denotes the performance drop, i.e., the accuracy difference between the first and the last sessions. "Final Improv." calculates the improvement of our method in the last session.

| Backbone | Methods | Accuracy in each session (%) | | | | | | | | | Avg. | PD | Final Improv. |
|---|---|---|---|---|---|---|---|---|---|---|---|---|---|
| | | 0 | 1 | 2 | 3 | 4 | 5 | 6 | 7 | 8 | | | |
| ResNet-18 | iCaRL (Rebuffi et al., 2017) | 64.10 | 53.28 | 41.69 | 34.13 | 27.93 | 25.06 | 20.41 | 15.48 | 13.73 | 32.87 | 50.37 | +44.81 |
| | TOPIC (Tao et al., 2020) | 64.10 | 55.88 | 47.07 | 45.16 | 40.11 | 36.38 | 33.96 | 31.55 | 29.37 | 42.62 | 34.73 | +29.17 |
| | ERL++ (Dong et al., 2021) | 73.62 | 68.22 | 65.14 | 61.84 | 58.35 | 55.54 | 52.51 | 50.16 | 48.23 | 59.29 | 25.39 | +10.31 |
| | F2M (Shi et al., 2021) | 71.45 | 68.10 | 64.43 | 60.80 | 57.76 | 55.26 | 53.53 | 51.57 | 49.35 | 59.14 | **22.10** | +9.19 |
| | Replay (Liu et al., 2022) | 74.40 | 70.20 | 66.54 | 62.51 | 59.71 | 56.58 | 54.52 | 52.39 | 50.14 | 60.77 | 24.26 | +8.40 |
| | ALICE (Peng et al., 2022) | 79.00 | 70.50 | 67.10 | 63.40 | 61.20 | 59.20 | 58.10 | 56.30 | 54.10 | 63.21 | 24.90 | +4.44 |
| | BiDist (Zhao et al., 2023) | 79.45 | 75.20 | 71.34 | 67.40 | 64.50 | 61.05 | 58.73 | 56.73 | 54.31 | 65.42 | 25.14 | +4.23 |
| | CEC+ (Wang et al., 2023) | 81.25 | 77.23 | 73.30 | 69.41 | 66.69 | 63.93 | 62.16 | 59.62 | 57.41 | 67.50 | 23.84 | +1.13 |
| | MetaAdapter (ours) | **82.00** | **78.34** | **74.13** | **70.33** | **67.09** | **64.24** | **62.67** | **60.16** | **58.54** | **68.61** | 23.46 | |
| ResNet-20 | CEC (Zhang et al., 2021) | 73.07 | 68.88 | 65.26 | 61.19 | 58.09 | 55.57 | 53.22 | 51.34 | 49.14 | 59.53 | **23.93** | +3.32 |
| | MetaFSCIL (Chi et al., 2022) | 74.50 | 70.10 | 66.84 | 62.77 | 59.48 | 56.52 | 54.36 | 52.56 | 49.97 | 60.79 | 24.53 | +2.49 |
| | FACT (Zhou et al., 2022a) | **78.83** | 72.71 | 68.63 | 64.71 | 61.48 | 58.34 | 56.00 | 53.85 | 51.84 | 62.93 | 26.99 | +0.62 |
| | TEEN (Wang et al., 2024) | 76.93 | 72.52 | **68.29** | 64.45 | 61.08 | 58.14 | 55.70 | 53.42 | 51.49 | 62.45 | 25.44 | +0.97 |
| | MetaAdapter (ours) | 76.83 | **72.60** | 68.28 | **64.90** | **61.83** | **59.34** | **57.60** | **55.78** | **52.46** | **63.29** | 24.37 | |
| ResNet-12 | NC-FSCIL (Yang et al., 2023) | 82.52 | 76.82 | 73.34 | 69.68 | 66.19 | 62.85 | 60.96 | 59.02 | 56.11 | 67.50 | 26.41 | +3.09 |
| | C-FSCIL (Hersche et al., 2022) | 77.47 | 72.40 | 67.47 | 63.25 | 59.84 | 56.95 | 54.42 | 52.47 | 50.47 | 61.64 | 26.99 | +8.73 |
| | OrCo (Ahmed et al., 2024) | 80.08 | 71.46 | 64.95 | 68.65 | 57.60 | 56.68 | 56.16 | 54.62 | 52.19 | 61.38 | 27.89 | +7.01 |
| | MetaAdapter (ours) | **84.05** | **78.86** | **75.16** | **71.64** | **68.29** | **65.31** | **63.54** | **61.52** | **59.20** | **69.73** | 24.85 | |

Table 6: Performance of FSCIL in each session on CUB200 and comparison with other methods. "Avg." is the average accuracy of all sessions. "PD" denotes the performance drop, i.e., the accuracy difference between the first and the last sessions. "Final Improv." calculates the improvement of our method in the last session.

| Backbone | Method | Accuracy in each session (%) | | | | | | | | | | | Avg. | PD | Final Improv. |
|---|---|---|---|---|---|---|---|---|---|---|---|---|---|---|---|
| | | 0 | 1 | 2 | 3 | 4 | 5 | 6 | 7 | 8 | 9 | 10 | | | |
| ResNet-18 | iCaRL (Rebuffi et al., 2017) | 68.68 | 52.65 | 48.61 | 44.16 | 36.62 | 29.52 | 27.83 | 26.26 | 24.01 | 21.16 | | 36.67 | 47.52 | +40.54 |
| | TOPIC (Tao et al., 2020) | 68.68 | 62.49 | 54.81 | 49.99 | 45.25 | 41.40 | 38.35 | 35.36 | 32.22 | 26.28 | | 43.92 | 42.40 | +35.42 |
| | ERL++ (Dong et al., 2021) | 73.52 | 71.09 | 66.13 | 63.25 | 59.49 | 58.89 | 58.64 | 57.72 | 56.15 | 52.28 | | 61.17 | 21.24 | +9.42 |
| | CEC (Zhang et al., 2021) | 75.85 | 71.94 | 68.50 | 63.50 | 62.43 | 58.27 | 57.73 | 55.81 | 54.83 | 52.28 | | 61.33 | 23.57 | +9.42 |
| | F2M (Shi et al., 2021) | 77.13 | 73.92 | 70.27 | 66.37 | 64.34 | 61.69 | 60.52 | 59.38 | 57.15 | 55.89 | | 63.96 | 21.24 | +5.81 |
| | Replay (Liu et al., 2022) | 75.90 | 72.14 | 68.64 | 63.76 | 62.58 | 59.11 | 57.82 | 55.89 | 54.92 | 52.39 | | 61.52 | 23.51 | +9.31 |
| | MetaFSCIL (Chi et al., 2022) | 75.90 | 72.41 | 68.78 | 64.78 | 62.96 | 59.99 | 58.30 | 56.85 | 54.78 | 52.64 | | 61.93 | 23.26 | +9.06 |
| | FACT (Zhou et al., 2022a) | 78.91 | 75.19 | 71.34 | 66.09 | 65.59 | 62.06 | 60.92 | 59.31 | 57.65 | 55.96 | | 64.55 | 22.95 | +5.74 |
| | TEEN (Wang et al., 2024) | 79.02 | 74.79 | 71.33 | 66.56 | 66.05 | 63.09 | 62.04 | 60.83 | 59.55 | 58.09 | | 65.48 | 20.93 | +3.61 |
| | BiDist (Zhao et al., 2023) | 79.12 | 74.99 | 70.87 | 67.30 | 65.89 | 63.45 | 61.40 | 60.11 | 58.61 | 57.48 | | 65.22 | 21.64 | +4.22 |
| | ALICE (Peng et al., 2022) | 77.40 | 72.70 | 70.60 | 67.20 | 65.59 | 63.40 | 62.90 | 61.90 | 60.50 | 60.10 | | 65.75 | **17.30** | +1.60 |
| | NC-FSCIL (Yang et al., 2023) | 80.45 | 75.98 | 72.30 | **70.28** | 68.17 | 65.16 | 64.43 | 63.25 | 60.66 | 59.44 | | 67.28 | 21.01 | +2.26 |
| | CEC+ (Wang et al., 2023) | 79.46 | 76.11 | 73.12 | 69.31 | 67.97 | 65.86 | 64.50 | 63.83 | 62.20 | 60.97 | | 67.76 | 18.49 | +0.73 |
| | OrCo (Ahmed et al., 2024) | 75.59 | 72.74 | 64.58 | 60.12 | 60.16 | 58.04 | 58.41 | 57.96 | 56.97 | 57.93 | | 61.86 | 17.66 | +3.77 |
| | MetaAdapter (ours) | **80.63** | **76.85** | **73.62** | 69.75 | **69.13** | **66.23** | **65.67** | **64.51** | **62.29** | **61.70** | | **68.45** | 18.93 | |

## C APPENDIX: VISUALIZATIONS OF EXPANDED FEATURES

In this part, we provide more analyses of our proposed MetaAdapter. In Figure 5, we visualize the feature embeddings and corresponding classification weights (*i.e.*, prototypes) from the mini-ImageNet test set, comparing the results using a frozen extractor and our proposed MetaAdapter after adaptation to novel classes. For clarity, we randomly select 5 base classes and 5 novel classes, with features from 100 test samples per class. As illustrated in the left part of Figure 5, the novel class prototypes tend to overlap and exhibit confusion with each other, with a less compact feature space when MetaAdapter is not applied. This occurs because the model, trained exclusively on base categories, fails to effectively adapt to novel concepts. In contrast, after applying MetaAdapter, these classes become more separable, and the feature space becomes more compact, as shown in the right part of Figure 5.

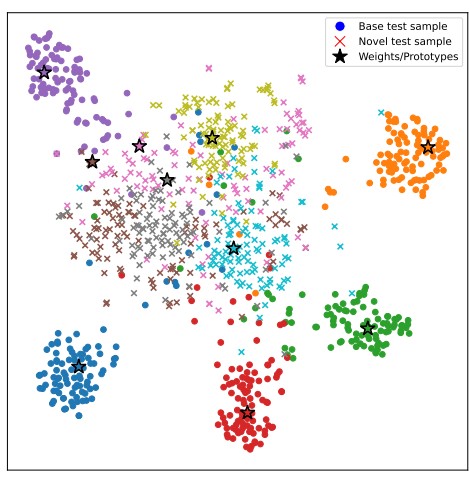 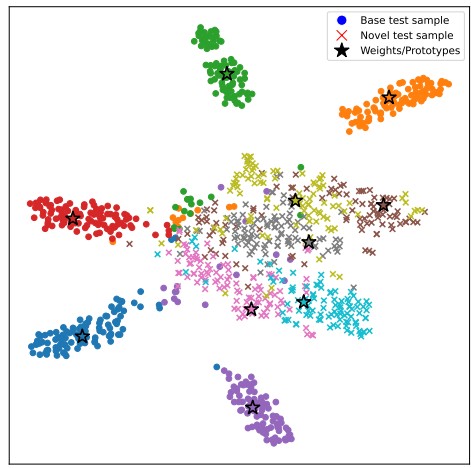

(a) Feature space with a frozen extractor      (b) Feature space with MetaAdapter

Figure 5: T-SNE (Van der Maaten & Hinton, 2008) plots of test samples and the corresponding classification weights/prototypes in the final session from mini-ImageNet with a frozen extractor (Wang et al., 2024) or our proposed MetaAdapter. Categories are represented by different colors. Best viewed in color.

## D  APPENDIX: ANALYSES OF INCREMENTAL SHOT

To further validate the effectiveness of our proposed method, we vary the shot number (*i.e.*, the number of training samples in each incremental class) in the original $N$-way $K$-shot few-shot class incremental learning task. As shown in Figure 6, our method demonstrates robustness even in extreme cases where only a single training sample (1-shot) is available. Moreover, as the number of training samples from novel classes increases, we observe corresponding performance improvements. This indicates that our approach can better adapt to incremental classes with additional training data, thereby proving the extendibility of our method.

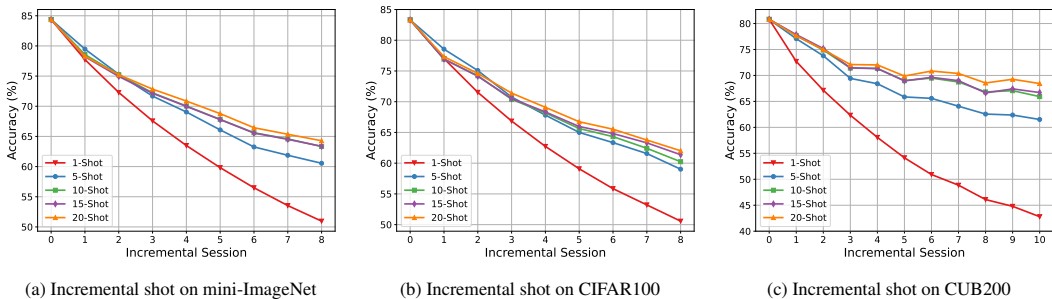

(a) Incremental shot on mini-ImageNet  (b) Incremental shot on CIFAR100  (c) Incremental shot on CUB200

Figure 6: Influence of different shot settings on incremental session accuracy.

## E  APPENDIX: ANALYSES OF CONFUSION MATRIX

To gain deeper insights into the specific challenges of the few-shot class incremental learning task, we present the confusion matrix results for (a) learning with afrozen extractor (Wang et al., 2024) and (b) MetaAdapter without FCL and (c) our full method in Figure 7.

We can see from Figure 7(a) that learning with a frozen extractor specializes in classifying base classes with concentrated values on the diagonal of these categories. However, it performs poorly on novel classes with much darker diagonal and and scattered prediction distribution, since the frozen extractor is only trained on the base training set without adaptation to novel classes.

Adapting with meta-initialized adapters in Figure 7(b) can better handle novel class samples, but struggles to sufficiently preserve base knowledge, resulting in a darker diagonal on base classes compared to Figure 7(c). It is because the severe data scarcity of few-shot class incremental learning not only causes the unique overfitting issue but also aggravates catastrophic forgetting.

As shown in Figure 7(c), with the proposed meta initialized adapters and feature compactness loss, our full method can address the above difficulties with concentrated values on the diagonal of both base and novel classes, confirming the observed performance gains in experiments.

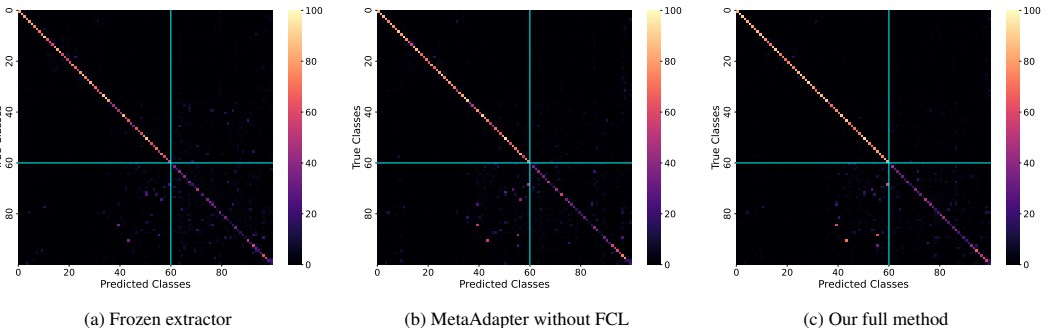

(a) Frozen extractor        (b) MetaAdapter without FCL        (c) Our full method

Figure 7: Confusion matrices of baseline approaches and our proposed method on the mini-ImageNet dataset are presented. The blue lines distinguish base and novel classes. Our method demonstrates significant improvements in prediction accuracy during the final session, as evidenced by a less scattered confusion matrix.

## F    APPENDIX: DATASETS OF DIFFERENT SEMANTIC SIMILARITIES

To provide a clearer understanding of the semantic characteristics of the benchmark datasets: mini-ImageNet (Russakovsky et al., 2015), CIFAR100 (Krizhevsky et al., 2009) and CUB200 (Wah et al., 2011), we describe the category information of the three datasets. The fine-grained classification dataset CUB200 consists solely of bird categories with similar appearances, leading to strong semantic correlations between the base and novel classes. It validates the empirical finding that more knowledge from base classes (*i.e.*, with a smaller value of the coefficient $w_a$ in Eq. (12) of the main paper) should be transferred for facilitating the learning of novel classes in CUB200 due to the strong semantic correlations between them. In contrast, images from the mini-ImageNet and CIFAR100 classification datasets exhibit more diverse visual appearances, with lower semantic similarity between base and novel classes compared to CUB200. The first half of the base classes in mini-ImageNet (indices 1-35) are animal-related (*i.e.*, 'house finch', 'robin', 'green mamba'), while the remaining base classes (indices 36-60) and the novel classes (indices 61-100) consist largely of inorganic objects. For CIFAR100, the dataset contains 100 classes grouped into 20 superclasses, covering a wide range of categories such as 'aquatic mammals', 'large carnivores' and 'household devices'. Similar to mini-ImageNet, the semantic differences between these categories are also significant, offering diverse visual representation across various domains. Thus, for better handling these classes, the model should pay more attention to the new few-shot classes by using a larger $w_a$, which further confirms the results of Figure 4(b) in our main paper.

