# OpenReview forum: "MetaAdapter: Leveraging Meta-Learning for Expandable Representation in Few-Shot Class Incremental Learning"
_ICLR.cc/2025/Conference — Submitted to ICLR 2025_

### Official Review · Reviewer_ma3t · 2024-10-22

**Soundness:** 2
**Presentation:** 2
**Contribution:** 1
**Rating:** 5
**Confidence:** 5

**Summary:**

This paper tackles the problem of Few-Shot Class Incremental Learning (FSCIL). The authors propose enhancing model plasticity during incremental learning stages by integrating and updating adapters within the backbone network. To simulate the testing scenario, the Reptile algorithm is employed to meta-learn the adapter, facilitating better initialization using data from the base session. Afterward, the adapters are frozen, and the backbone is fine-tuned on the base classes using a novel Feature Compactness Loss (FCL), complemented by a strategy to promote Flat Local Minima (FLM). The objectives of FCL and FLM are to reduce inter-class distances within the base classes, thereby preserving feature space capacity for future incremental classes. During incremental sessions, the adapters are updated and merged into the backbone through a running average. The approach demonstrates superior performance across three standard benchmarks.

**Strengths:**

1.	Enhancing model plasticity through the insertion of adapters while maintaining stability by freezing the backbone is a sound technical approach for FSCIL.
2.	Preserving feature space for future incremental classes is an effective strategy for improving overall performance.
3.	The proposed method demonstrates superior performance across three standard benchmarks.

**Weaknesses:**

Major:

1. The primary motivation of this paper is that previous methods do not update the learned representation during incremental sessions, thereby compromising model plasticity. However, the authors have not sufficiently explored related work in the field of continual learning. There is a considerable body of research that focuses on balancing model stability and plasticity during incremental sessions while also updating the backbone, not limited to FSCIL. The absence of a discussion on these relevant works is a notable gap.
Even within FSCIL, prior works such as MetaFSCIL provide a relevant comparison. MetaFSCIL is a meta-learning-based method that not only learns meta-representations using base session data but also updates the backbone representation during incremental sessions. The meta-learning strategy employed in offline training mimics the meta-testing scenario while also balancing stability and plasticity as the backbone is updated. Conceptually, MetaFSCIL is closely aligned with the idea of meta-learning adapters proposed in this paper. However, the authors have misinterpreted MetaFSCIL’s approach in L60-63 and L113-115.
2. The concept of feature compactness loss, aimed at reducing excessive dispersion among base classes, is conceptually similar to the forward compatibility strategy introduced in FACT, which ensures sufficient feature space is reserved for future classes. However, the authors did not provide a discussion or comparison with FACT, despite the conceptual overlap. Including such a comparison would have strengthened the paper’s positioning and clarified its contributions relative to existing approaches.

3. The training sequence, where adapters are trained before fine-tuning the backbone, appears non-intuitive. In the first phase, the adapters are trained while the backbone remains frozen, causing the adapters to rely heavily on the backbone’s fixed knowledge. As a result, the adapters are meta-learned to operate on top of this frozen representation. However, in the second phase, the backbone undergoes fine-tuning, altering its parameters. This shift may create incompatibility between the previously trained adapters and the newly updated backbone, potentially undermining the synergy between the two components.


Minor:

1.	It is inaccurate to state “randomly initialize the adapter parameters for the j-th task,” as mentioned in L197-198. This phrasing implies that the adapters are randomly initialized for each task, which is misleading. In reality, the adapters should be updated iteratively using Eqs. (1) and (2) to build on previously learned knowledge rather than restarting with random parameters for every task.
2.	Eq. (4) appears somewhat unclear. If the goal is to bring feature vectors closer together, it would imply that Eq. (4) encourages a more uniform probability distribution. However, it is unclear how the information is concatenated into P_{\text{concat}} . What is the dimensionality of P_{\text{concat} ? If the concatenation occurs along the embedding dimension, the resulting output of the cosine similarity would be a scalar. Applying softmax to a scalar value does not seem meaningful, so additional clarification on the concatenation process is needed.
3.	The objective of the feature compactness loss and sharpness-aware minimization is to reduce the distances among base classes. However, it is unclear whether this operation could negatively impact the model’s performance on the base classes. If such degradation occurs, it is important to discuss how this issue could be mitigated to maintain performance on the base classes.

**Questions:**

1. How is the knowledge encoded in the adapters ensured to be task-agnostic, as claimed in L67? This concept is introduced in the paper’s introduction but is not elaborated upon in subsequent sections. A more detailed explanation is necessary to clarify how the adapters generalize across tasks without being biased toward specific ones.
2. How are pseudo-targets generated? The paper lacks details on the process used to obtain these pseudo-targets. Providing a clear description of the method for generating pseudo-targets is essential for understanding the approach and evaluating its effectiveness.

---

> ### Author Response · Authors · 2024-11-22
> **Response to Reviewer ma3t (1/2)**
>
> Thanks for your constructive comments! We hope our responses address your concerns, and we kindly invite you to reassess our submission. Please let us know if you have any further questions or suggestions.
>
> ---
>
> **Q1**: The primary motivation of this paper is that previous methods do not update the learned representation during incremental sessions, thereby compromising model plasticity. However, there is a considerable body of research that focuses on balancing model stability and plasticity during incremental sessions while also updating the backbone. This broader body of work is not sufficiently discussed.
>
> **A1**: In the FSCIL setting,  continual updates often result in significant performance drops for the base session due to the limited data available for new classes. This is why many existing methods in FSCIL prioritize stability over continuous backbone adaptation [1]. Our approach focuses on addressing this challenge by using lightweight meta-initialized adapters to enhance flexibility while preserving stability, without overly compromising the performance on base classes.
>
> ---
>
> **Q2**: MetaFSCIL, a method that updates the backbone during incremental sessions using meta-learning, aligns conceptually with the approach proposed in this paper. However, this method is not adequately discussed, and its relevance is under explored.
>
> **A2**:  Thank you for your valuable suggestion. We agree that MetaFSCIL is an important related method, and we will include a more detailed discussion in the revised manuscript to highlight its relevance and differentiate it from our approach. To clarify, MetaFSCIL samples a sequence of sessions to mimic the evaluation protocol during the base phase and evaluates the  model using a meta-objective. In contrast, our MetaAdapter approach is designed with a different focus. Instead of mimicking meta-testing during the base phase, we focus on using meta-learning to obtain meta-initialized adapters that provide a generalizable starting point for expanding and refining feature representations.  During the online incremental learning stage, MetaFSCIL uses Bi-directional Guided Modulation (BGM) to generate activation masks to mitigate forgetting. In comparison, our MetaAdapter framework keeps the backbone frozen and utilize it as a teacher model for knowledge distillation to guide the adaptation of lightweight adapters.
>
> ---
>
> **Q3**: The concept of feature compactness loss (FCL) seems to overlap with FACT. A comparison with FACT would strengthen the contribution.
>
> **A3**: FACT uses manifold mixup during the base session to generate virtual classes, which creates space for new categories in subsequent incremental learning stages. Our feature compactness loss (FCL) takes a different approach by compacting both inter-class and intra-class distances during the base session. This design prevents the embedding space from becoming overly dispersed, effectively enabling the model to better adapt to new tasks in future incremental sessions.
>
> To further substantiate the advantages of FCL, we conducted additional experiments in the **General Response** to analyze the relationship between inter-class feature representations and embedding space reservation. Specifically, we evaluated the **relative angular disparity** $T(f_\theta)$ between new-class samples and base-class prototypes (Table 3) and the **inter-class angular distance** among base-class prototypes (Table 4). The results in Table 4 show that Inter-class angular distance among base-class prototypes decreases as $w_{fcl}$ increases, which shows that FCL effectively compacts the embedding space. Meanwhile, The $T(f_\theta)$ metric in Table 3 improves when $w_{fcl}$ is set to a moderate value, showing that reducing the angular separation among base-class features to an appropriate extent supports better adaptation to new classes.
>
> ------
>
> **Q4**: The training sequence appears non-intuitive. Adapters are trained before fine-tuning the backbone, which may create inconsistencies.
>
> **A4**: Thank you for this observation. The reason we train adapters before fine-tuning the backbone is due to the nature of the meta-learning task. In our approach, the meta-learning phase focuses on tasks involving unseen categories. If we fine-tune the backbone first, it would have already seen all categories, which would defeat the purpose of meta-learning by reducing its ability to generalize to new, unseen classes.
>
> ---

---

> > ### Author Response · Authors · 2024-11-22
> > **Response to Reviewer ma3t (2/2)**
> >
> > **Q5**: It is inaccurate to state “randomly initialize the adapter parameters for the j-th task,” as mentioned in L197-198.
> >
> > **A5**: Thank you for pointing this out. We agree that our statement in Lines 197-198 was unclear. The adapter parameters $\theta_a$ are initialized once with a shared random initialization for all tasks, rather than separately for each task. We have revised the manuscript to clarify this.
> >
> > ------
> >
> > **Q6**: Equation (4) and the dimensionality of $P_{concat}$ are unclear, especially regarding the application of softmax.
> >
> > **A6**: To clarify, 'c_pseudo' represents prototypes for classes not present in the current batch, derived from the mean feature vectors of these classes from the previous epoch. The batch means 'c_batch', are prototypes for each class in the current batch, and $\mathbf{p}$ represents the feature vectors of all samples in the current batch. For example, if 'c_batch' has a shape of (N, d), 'c_pseudo' has a shape of (K, d), and $\mathbf{p}$  has a shape of (M, d), then 'p_concat' will have a shape of ((N + K + M), d) after concatenation. Regarding the cosine similarity, we compute it between all pairs of vectors in 'p_concat', which results in a similarity matrix of shape ((N + K + M), (N + K + M)). The softmax is applied row-wise to these cosine similarities (excluding self-similarities on the diagonal) to convert them into a probability distribution, which is then used in our loss function.
> >
> > ---
> >
> > **Q7**: The objective of FCL is to reduce the distances among base classes, but it is unclear if this impacts performance on base classes.
> >
> > **A7**: As shown in the **General Response**, we conducted experiments on base accuracy across varying  $w_{fcl}$ values (Table 1). As can be seen from Table 1, the base accuracy is relatively stable across varying $w_{fcl}$ values for all datasets. However, excessively high values (e.g., $w_{fcl}= 3.0$) lead to a slight drop in performance due to over-compactness in the embedding space, which can harm the representation of base classes.
> >
> > ---
> >
> > **Q8**: How is the knowledge encoded in adapters ensured to be task-agnostic, as claimed in L67?
> >
> > **A8**: As explained in **Section 3.2**, we achieve task-agnostic knowledge by using meta-learning. In the first phase, we create few-shot tasks by randomly sampling instances from base classes and then use the Reptile algorithm to train the adapters. This approach helps the adapters acquire generalizable parameters that can quickly adapt to new tasks.
> >
> > ---
> >
> > **Q9**: The process for generating pseudo-targets is unclear.
> >
> > **A9**: We would like to clarify that the process of generating pseudo-targets is explained in detail in **Section 3.2** of the manuscript. Briefly, we partition the base label space $Y_0$ into non-overlapping subsets $ \hat{Y}_1, \hat{Y}_2, \ldots, \hat{Y}_C $. For each subset $ \hat{Y}_i $, we randomly sample $ \hat{K} $ examples to form an $ \hat{N} $-way, $ \hat{K} $-shot support set $ \mathcal{S}^i $. These support sets $ \mathcal{S}^1, \mathcal{S}^2, \ldots, \mathcal{S}^C $ are then used to train the adapters using the Reptile algorithm. This approach ensures consistency with the few-shot learning setup and enables efficient task adaptation.
> >
> > ------
> >
> > [1] A Survey on Few-Shot Class-Incremental Learning. Neural Networks 2024.

---

> > > ### Comment · Reviewer_ma3t · 2024-11-26
> > > **Follow-up comments for the Authors**
> > >
> > > Thank you for your detailed response. While I appreciate the effort, some of my concerns remain only partially addressed. I would like to highlight a few points for further clarification:
> > >
> > > 1. Plasticity vs. Base Performance
> > > In your response, you mention that the proposed method improves the tradeoff between stability and plasticity by freezing the backbone. However, it has not been experimentally validated that the proposed method demonstrates better plasticity. Based on the reported results in Table 1 and Table 5, the improvements observed in terms of (base, last, and average performance) (+12.08, +11.82, +11.9) and (+9.55, +9.23, +8.94) over MetaFSCIL seem primarily driven by better performance on the base classes. This suggests that the gains may not be related to improved plasticity, but rather to the bias introduced by the base classes, as pointed out by Reviewer 6ye5. I believe this discrepancy needs further clarification.
> > >
> > > 2. Comparison with Prior Work
> > > As noted in the initial review, the comparison with relevant prior works is not fully incorporated in the revised version of the paper. Without this, it is challenging to assess how your approach fits within the existing body of research. I encourage you to include a more comprehensive discussion of prior work to strengthen the paper's context.
> > >
> > > 3. Training Sequence and Knowledge Misalignment
> > > Regarding the training sequence, I remain unconvinced by the response. The pre-training data (e.g., ImageNet) may already include base classes or similar categories to those found in datasets like Mini-ImageNet or CIFAR100. Consequently, the first step cannot be entirely focused on training unseen classes, as this could introduce knowledge misalignment between the adapters and the representations. This concern, which I raised in my initial review, has not been fully addressed. I believe it would be helpful for the authors to provide further justification or clarity on this point.
> > >
> > > 4. Pseudo-Targets and Confusion in Terminology
> > > There seems to be some confusion regarding the pseudo-targets concept. As mentioned in Sec. 3.3 (rather than Sec. 3.2), I would like to clarify that my comment referred to the pseudo-targets for unseen categories, which are derived from the average feature representations computed in the previous epoch. If we are treating the current batch as unseen, I am unclear how the pseudo-targets from the previous epoch can be determined in this context. Further explanation would be helpful here.
> > >
> > > 5. Fairness with Different Backbones
> > > The discrepancy between different backbones (ResNet-18 and ResNet-12) in terms of fairness, as pointed out by Reviewer 6ye5, has not been adequately addressed. I strongly encourage the authors to carefully consider this issue and revise the paper accordingly, either in this submission or in future iterations.
> > >
> > > If these concerns can be thoroughly addressed, I would be happy to reconsider the paper's evaluation.

---

> > > > ### Author Response · Authors · 2024-11-29
> > > > **Response to Reviewer ma3t**
> > > >
> > > > Thanks for your constructive comments! Below, we address your concerns point by point.
> > > >
> > > > ------
> > > >
> > > > **Q1**: The proposed method improves the trade-off between stability and plasticity by freezing the backbone, but there is no experimental validation of improved plasticity. Based on the results in Table 1 and Table 5, the improvements seem primarily driven by better performance on base classes, potentially due to bias introduced by the base classes.
> > > >
> > > > **A1**: Thank you for raising this point. As shown in Table 3 of the manuscript and Tables 1 and 2 of the General Response, our method demonstrates superior performance in both base accuracy and incremental accuracy compared to NC-FSCIL [1]. This indicates that the observed improvements are not solely due to the strong performance on base classes but also reflect better adaptability to incremental sessions. To provide a clearer view of stability across sessions, we also include the performance drop (PD) metric in Table 1,5 and 6 in the revised manuscript. For mini-ImageNet, the PD results show that our method outperforms other methods on ResNet-12 and is comparable to recent methods on ResNet-18. For CIFAR100, the PD results indicate that we achieve superior performance on ResNet-12 and remain comparable to other methods on ResNet-18 and ResNet-20. These findings demonstrate that our approach effectively enhances plasticity while maintaining competitive stability.
> > > >
> > > > ------
> > > >
> > > > **Q2**: The revised version lacks a comprehensive discussion of relevant prior works, which makes it difficult to assess how the proposed method fits within the existing research landscape.
> > > >
> > > > **A2**: Thank you for the suggestion. In the revised manuscript, we include a more detailed discussion of prior works, focusing on dynamic neural networks and their strategies for enhancing plasticity.  As these approaches often involve increased architectural complexity, which can reduce efficiency, we propose using lightweight, meta-initialized adapters, which allow the model to efficiently adapt to new few-shot tasks without significantly increasing the model's complexity. Additionally, we also provide a detailed comparison with MetaFSCIL to highlight the key differences of our method.
> > > >
> > > > ------
> > > >
> > > > **Q3**: The use of pre-training data (e.g., ImageNet) may already include base classes or similar categories. Consequently, the first step cannot be entirely focused on training unseen classes.
> > > >
> > > > **A3**:  Thank you for raising this concern. For mini-ImageNet and CIFAR100, we train the model from scratch without pre-training on ImageNet. Instead, the model is pre-trained on half of the base classes to initialize the feature extractor, while the remaining half of the classes are used for meta-training the adapters. These details are provided in Appendix A of the manuscript.
> > > >
> > > > ------
> > > >
> > > > **Q4**: There is confusion regarding pseudo-targets for unseen categories derived from the previous epoch.
> > > >
> > > > **A4**: Thank you for raising this point. We would like to clarify that the pseudo-targets refer to the mean features of unseen classes within the current batch.  These mean features are computed at the end of the previous epoch. These details have been included in Section 3.4 of  the revised manuscript.
> > > >
> > > > ------
> > > >
> > > > **Q5**: The use of different backbones raises concerns about the fairness of comparisons.
> > > >
> > > > **A5**: Thank you for raising this concern. To ensure fairness, we provide additional results using ResNet-18 for mini-ImageNet,as shown in Table 1, and ResNet-18 and ResNet-20 for CIFAR100, detailed in Table 5 of the revised manuscript. These results demonstrate that our method improves both the final accuracy and average accuracy on the same backbone compared to other methods.
> > > >
> > > > ------
> > > >
> > > > [1] Neural Collapse Inspired Feature-Classifier Alignment for Few-Shot Class-Incremental Learning. ICLR2023.

---

> > > > > ### Author Response · Authors · 2024-12-02
> > > > > **To Reviewer ma3t**
> > > > >
> > > > > Dear Reviewer  ma3t,
> > > > >
> > > > > Thank you for your careful review and thoughtful feedback! We have updated our submission based on your suggestions and provided detailed responses to the newly raised concerns. As the discussion phase is nearing its conclusion, could you please check our response to see if we address your concerns? We are happy to address further questions you might have.
> > > > >
> > > > > Best regards,
> > > > >
> > > > > Authors

---

> > > > > > ### Comment · Reviewer_ma3t · 2024-12-02
> > > > > > **Final response to authors**
> > > > > >
> > > > > > Thank you for your response. While my concerns have been partially addressed, I’ve decided to raise my score to 5, which is just below the acceptance threshold. Here are the main reasons I can’t give a positive rating:
> > > > > >
> > > > > > 1. Limited novelty: The paper’s contribution seems somewhat incremental, as the motivation behind meta-learning and feature compactness overlaps with existing work.
> > > > > >
> > > > > > 2. Effectiveness of the proposed method: I’m still not fully convinced by the method’s effectiveness. The benchmark results show some performance issues, especially with the forgetting problem, which is a key challenge in continual learning. Additionally, the method seems to rely heavily on achieving high accuracy on the base classes.
> > > > > >
> > > > > > 3. Training sequence concerns: The response for the training sequence cannot fully convince me. The ability to learn incrementally depends on what knowledge is already learned. I understand that the authors aim to mimic the process of learning new classes, but there’s an issue with the adapters. Since the adapters are attached to the backbone, any new knowledge learned by the adapters is also tied to the backbone. In the third stage, when the backbone is retrained on the full set of base classes, the adapters are frozen, which could lead to inconsistencies in how knowledge is updated. It’s also worth mentioning that in regular few-shot learning, pre-training on the full base classes, including during meta-learning—has been shown to be very effective (Meta-Baseline: Exploring Simple Meta-Learning for Few-Shot Learning, ICCV 2021).
> > > > > >
> > > > > > Given these points, I’m leaning toward recommending rejection.

---

> > > > > > > ### Author Response · Authors · 2024-12-03
> > > > > > > **Response to Reviewer ma3t**
> > > > > > >
> > > > > > > Thank you for your constructive feedback and for raising the score of our submission! We highly appreciate your time and consideration. Below, we address your concerns point by point.
> > > > > > >
> > > > > > > ------
> > > > > > >
> > > > > > > **Q1**: The paper’s contribution seems somewhat incremental, as the motivation behind meta-learning and feature compactness overlaps with existing work.
> > > > > > >
> > > > > > > **A1**: Thank you for raising this point. While our work builds on existing concepts, we address critical limitations that have not been fully explored in prior studies.
> > > > > > >
> > > > > > > The motivation behind the Feature Compactness Loss (FCL) stems from the limitations of traditional pre-training methods, which often optimize empirical loss and maximize inter-class margins for base-class prototypes. While these strategies enhance feature discrimination, they may result in overfitting to base classes and reduced adaptability to few-shot new classes. Additionally, this strategy overlooks the issue of Minority Collapse [1], where few-shot class features cluster too tightly in imbalanced scenarios, potentially resulting in performance degradation for new classes. To mitigate these issues and reserve capacity for incremental learning, we propose the Feature Compactness Loss (FCL). By compacting both inter-class and intra-class distances, FCL prevents the embedding space from becoming overly dispersed, which preserves learning capacity for future few-shot incremental learning scenario.
> > > > > > >
> > > > > > > Regarding the motivation for meta-learning, unlike prior works that simulate incremental tasks using base-class data, we leverages meta-learning to produce meta-initialized adapters. These adapters are encoded with task-agnostic knowledge and provide a generalizable starting point to refine feature representations.
> > > > > > >
> > > > > > > ------
> > > > > > >
> > > > > > > **Q2**: The effectiveness of the proposed method remains unclear. The benchmark results show some performance issues, especially regarding forgetting, which is a key challenge in continual learning. Additionally, the method seems to rely heavily on achieving high accuracy on the base classes.
> > > > > > >
> > > > > > > **A2**: Thank you for raising this concern. Our method effectively mitigates forgetting, as shown by the performance drop (PD) metric in Table 1, 5, and 6 of the revised manuscript. While the PD metric reflects a model's ability to mitigate forgetting, it represents only one aspect of performance. For example, a model with consistently 0 accuracy across sessions would have a PD of 0, which might misleadingly appear optimal. Thus, PD must be evaluated alongside accuracy for a comprehensive assessment.
> > > > > > >
> > > > > > > Moreover, the results in Table 3 of the manuscript and Tables 1 and 2 of the General Response highlight that our method achieves higher base accuracy and incremental accuracy compared to NC-FSCIL [2].  This shows that the observed improvements are not merely due to strong performance on base classes but also stem from enhanced adaptability during incremental sessions.
> > > > > > >
> > > > > > > ------
> > > > > > >
> > > > > > > **Q3**: Concerns about the training sequence remain. Additionally, pre-training on the full base classes, as shown in Meta-Baseline (ICCV 2021), has been proven effective in regular few-shot learning.
> > > > > > >
> > > > > > > **A3**:  Thank you for raising this concern. The reason why we train adapters before fine-tuning the backbone  is to preserve the meta-learning phase's focus on unseen categories. We also acknowledge that pre-training on the full base classes, as demonstrated in Meta-Baseline (ICCV 2021), is effective in regular few-shot learning. However, such methods typically rely on a frozen feature extractor, which can limit plasticity. In contrast, our approach uses meta-initialized adapters to reduce overfitting while enhancing the model's plasticity in the FSCIL scenario.
> > > > > > >
> > > > > > > ------
> > > > > > >
> > > > > > > [1] Neural collapse inspired attraction--repulsion-balanced loss for imbalanced learning. Neurocomputing2023.
> > > > > > >
> > > > > > > [2] Neural Collapse Inspired Feature-Classifier Alignment for Few-Shot Class-Incremental Learning. ICLR2023.

---

### Official Review · Reviewer_6ye5 · 2024-10-25

**Soundness:** 2
**Presentation:** 2
**Contribution:** 2
**Rating:** 5
**Confidence:** 5

**Summary:**

Pointing out the heavy reliance on a feature extractor trained only on the base session, this paper leverages meta-learning to effectively adapt to new classes. Specifically, the proposed method first constructs a meta-learning scenario using the dataset from the base session and trains an adapter with Reptile, one of the meta-learning algorithms. Then, MetaAdapter trains a backbone network with a feature compactness loss to reserve feature space for future new classes. Finally, MetaAdapter updates the adapter using few-shot new-class data. The authors evaluate the proposed method on the CIFAR-100, miniImageNet, and CUB-200 datasets.

**Strengths:**

The authors address one of the key challenges in Few-Shot Class Incremental Learning (FSCIL): the lack of plasticity caused by the heavy reliance on the encoder trained during the base session. To overcome this, they propose leveraging a meta-learning approach and tackle several challenges that arise when applying meta-learning in the context of FSCIL.

**Weaknesses:**

1) Unclear descriptions of the proposed method.

The meaning of 'c_pseudo,' mentioned in lines L223-L224, is difficult to understand, and a formal definition would be helpful. Additionally, in Equation 4, the dimension of 'p_concat' is unclear. It appears that 'c_batch' has a shape of B x C x d, while 'p' has a shape of B x d, where B, C, and d represent the batch size, the number of base classes, and the feature dimension, respectively. If this is the case, concatenation would be impossible; if not, further clarification from the authors on the structure of 'p_concat' is necessary.

Furthermore, Section 3.5 is challenging to interpret. Figure 2 is particularly difficult to follow, especially in relation to the adapter’s structure. It seems that the adapter may share convolutional layers with the backbone model, as indicated by the gray and sky-blue colors. However, the gray coloring appears to make this unclear. Additionally, the number of channels between the adapter convolutional layer (shown in red) and the backbone convolutional layer (in sky-blue) seems to differ. Yet, in Equation 13, these two layers are simply added, which would not be feasible with different channel numbers.

These issues make it challenging to understand the few-shot adaptation phase. A more explicit explanation of the adapter architectures would be beneficial.

2) Motivation of the feature compactness loss (FCL)

In L211-L215, the authors argue that traditional optimization during the base session results in a dispersed embedding space that does not accommodate future new classes. To address this, they propose Feature Compactness Loss (FCL), which compacts the feature space to reserve space for future new classes.
With similar motivation, many existing works on Few-Shot Class Incremental Learning (FSCIL) [1, 2] have aimed to maximize the margin between classes in the feature space. While FCL appears to share this motivation, it takes the opposite approach: rather than maximizing the margin, it reduces the overall feature space. To validate FCL, the authors should provide additional analysis to explain why compacting the feature space is more effective than maximizing class margins in preserving space for new classes.
The reviewer encourages the authors to refer to [3], which proposes reducing inter-class distance to improve representation learning in FSCIL and provides an analysis on its implications.

[1] Yang et al, "Neural collapse inspired feature-classifier alignment for few-shot class incremental learning", in ICLR 2023.

[2] Zhou et al, "Forward compatible few-shot class-incremental learning", in CVPR2022.

[3] Oh et al, "CLOSER: Towards Better Representation Learning for Few-Shot Class-Incremental Learning", in ECCV2024.

3) Fairness issue

In Appendix A, the authors state that they use ResNet-12 for both mini-Imagenet and CIFAR-100 experiments.
However, several existing methods like FACT and ALICE adopt ResNet-18 for miniImageNet experiments.
Thus, the comparison with these methods is unfair and may not demonstrate the effectiveness of the proposed method.

**Questions:**

Please refer to the weakness part.

---

> ### Author Response · Authors · 2024-11-22
> **Response to Reviewer 6ye5 (1/2)**
>
> Thanks for your constructive comments! We hope our responses address your concerns, and we kindly invite you to reassess our submission. Please let us know if you have any further questions or suggestions.
>
> ---
>
> **Q1**: The meaning of 'c_pseudo,' mentioned in lines L223-L224, is difficult to understand, and a formal definition would be helpful. Additionally, in Equation 4, the dimension of 'p_concat' is unclear. It appears that 'c_batch' has a shape of $B \times C \times d $, while 'p' has a shape of $B \times d$. If this is the case, concatenation would be impossible; if not, further clarification on the structure of 'p_concat' is necessary.
>
> **A1**: We appreciate your suggestion and will provide a formal definition of 'c_pseudo' and clarify the dimensions of 'p_concat' in the revised version of our manuscript. To clarify, 'c_pseudo' refers to prototypes for categories not present in the current batch, derived from the mean feature vectors of these unseen classes from the previous epoch. The batch means $\mathbf{c}_{\text{batch}}$ are prototypes for each class in the current batch, while the original feature vectors $\mathbf{p}$ represent the features of all samples in the current batch. We concatenate these three components to form 'p_concat'. For example, if 'c_batch' has a shape of (N, d), 'c_pseudo' has a shape of (K, d), and $\mathbf{p}$  has a shape of (M, d), then 'p_concat' will have a shape of ((N + K + M), d) after concatenation.
>
>
> ---
>
> **Q2**:  Figure 2 is difficult to follow, particularly regarding the adapter’s structure. The adapter appears to share convolutional layers with the backbone model, as suggested by the gray and sky-blue colors. However, the gray coloring makes this unclear.
>
> **A2:** To clarify, the gray components in Figure 2 represent the convolutional layers in the backbone where no adapter is applied. These layers are shared across all tasks and remain unchanged during incremental sessions. The sky-blue components indicate the convolutional layers within the blocks that contain parallel adapters. These adapters form residual connections with the corresponding convolutional layers in the backbone, enabling task-specific adaptation without modifying the structure of the backbone.
>
> ------
>
> **Q3**:  The channel dimensions between the adapter and backbone layers seems to differ, but Equation 13 suggests a straightforward addition, which seems inconsistent if the channel numbers do not match.
>
> **A3**:  The addition of the adapter and backbone layers is done by merging the final $3 \times 3$ kernel with the $1 \times 1$ kernels. We achieve this by zero-padding the $1 \times 1$ kernels and aligning them at the center of the $3 \times 3$ kernel. This transformation requires both layers to have the same stride, with the $1 \times 1$ layer having one pixel less padding. For example, if the $3 \times 3$ layer uses padding = 1 (commonly used), the $1 \times 1$ layer should have padding = 0, ensuring consistent channel dimensions for addition.
>
> ---

---

> ### Author Response · Authors · 2024-11-22
> **Response to Reviewer 6ye5 (2/2)**
>
> **Q4**: The motivation for Feature Compactness Loss (FCL) is to compact the feature space after the base session to reserve room for new classes. However, existing FSCIL methods with similar motivations focus on maximizing inter-class margins instead. Given this contrast, additional analysis is needed to explain why compacting features is more effective than maximizing margins.
>
> **A4:** Existing FSCIL methods that focus on maximizing inter-class margins, such as ALICE [1], aim to enhance feature discrimination by increasing the separation between base-class prototypes. However, as CLOM [2] suggests, this strategy can lead to overfitting on base classes, which may result in degraded generalization to few-shot new classes. Moreover, such methods ignores the issue of **Minority Collapse [3]** , where features of few-shot classes cluster excessively tightly in imbalanced scenarios, further reducing the model's adaptability to new classes. In contrast, our Feature Compactness Loss (FCL) is specifically designed to compact both inter-class and intra-class distances during the base session. This design prevents the embedding space from becoming overly dispersed, effectively enabling the model to better adapt to new tasks in future incremental sessions. As demonstrated by the experimental results in our manuscript, our method outperforms ALICE.
>
> To further substantiate the advantages of FCL, we conducted additional experiments in the **General Response** to analyze the relationship between inter-class feature representations and embedding space reservation. Specifically, we evaluated the **relative angular disparity** $T(f_\theta)$ between new-class samples and base-class prototypes (Table 3) and the **inter-class angular distance** among base-class prototypes (Table 4). The results in Table 4 show that Inter-class angular distance among base-class prototypes decreases as $w_{fcl}$ increases, which shows that FCL effectively compacts the embedding space. Meanwhile, The $T(f_\theta)$ metric in Table 3 improves when $w_{fcl}$ is set to a moderate value, showing that reducing the angular separation among base-class features to an appropriate extent supports better adaptation to new classes.
>
> ---
>
> **Q5**: Fairness issue: In Appendix A, you use ResNet-12 for both mini-ImageNet and CIFAR-100 experiments, while other methods like FACT and ALICE adopt ResNet-18 for mini-ImageNet. This discrepancy may result in unfair comparisons.
>
> **A5**: In our experiments, we observed that prior studies use different backbone architectures for mini-ImageNet and CIFAR100, as highlighted in Table 8 of our paper. We followed C-FSCIL [4] and selected the shallowest architecture, ResNet-12, for both mini-ImageNet and CIFAR100. For CUB200, all methods consistently use the same architecture, ResNet-18 pretrained on ImageNet.
>
> Additionally, the parameter differences between ResNet-12 and ResNet-18 are relatively small. Specifically:
>
> - **ResNet-18**: 11.57M total parameters, with 0.29M trainable parameters for adapters.
> - **ResNet-12**: 12.81M total parameters, with 0.32M trainable parameters for adapters.
>
> ------
>
> [1] Few-Shot Class-Incremental Learning from an Open-Set Perspective.  ECCV2022.
>
> [2] Margin-based few-shot classincremental learning with class-level overfitting mitigation. NeurIPS2022.
>
> [3] Neural collapse inspired attraction--repulsion-balanced loss for imbalanced learning. Neurocomputing2023.
>
> [4] Constrained Few-shot Class-incremental Learning. CVPR2022.

---

> > ### Comment · Reviewer_6ye5 · 2024-11-25
> >
> > Dear Authors,
> >
> > Thank you for your effort and detailed responses.
> > Your rebuttal has partially addressed my concerns, including the motivation for the FCL loss.
> > However, I think the issue of unclear descriptions has not been sufficiently addressed.
> > While your responses have clarified my questions and improved my understanding, a paper should be written in a way that allows readers to comprehend it clearly without requiring additional explanation.
> > I kindly request that you revise the paper to ensure it is self-explanatory and easy to follow. Once revised, I will be happy to reconsider whether the updated version resolves these concerns.
> >
> > Additionally, the fairness issue remains inadequately addressed.
> > I am aware that ResNet-12 often outperforms ResNet-18 despite the small differences. As such, comparing results from ResNet-12 with those from ResNet-18 is both unfair and meaningless.
> > Moreover, the superior results of the proposed method in the last session seem to stem largely from its strong performance in the base session.
> > To account for this, it would be better to include an additional metric, such as performance drop (PD) across incremental sessions. Based on PD, I find the proposed method may not be the most effective.
> > Although this critique was not included in my original review, and I will not factor it into my rating, I encourage you to consider this aspect in your revisions.

---

> > > ### Author Response · Authors · 2024-11-29
> > > **Response to Reviewer 6ye5**
> > >
> > > Thanks for your constructive comments! Below, we address your concerns point by point.
> > >
> > > ------
> > >
> > > **Q1**: The paper should be written in a way that is self-explanatory and easy to follow without requiring additional explanation.
> > >
> > > **A1**: Thank you for the suggestion. In the revised manuscript, we elaborate on the motivation behind the Feature Compactness Loss (FCL) and provide a clearer explanation of its implementation.
> > >
> > > ------
> > >
> > > **Q2**: The fairness issue remains inadequately addressed.
> > >
> > > **A2**: Thank you for raising this concern. To ensure fairness, we provide additional results using ResNet-18 for mini-ImageNet, as shown in Table 1, and ResNet-18 and ResNet-20 for CIFAR100, detailed in Table 5 of the revised manuscript. These results demonstrate that our method improves both the final accuracy and average accuracy on the same backbone compared to other methods.
> > >
> > > ------
> > >
> > > **Q3**:  The superior results in the last session appear to rely heavily on the strong performance in the base session. It would be better to include an additional metric, such as performance drop (PD) across incremental sessions.
> > >
> > > **A3**:  Thank you for raising this concern. NC-FSCIL [1] is a strong recent baseline, and we observe consistent improvements with our method across multiple datasets. On mini-ImageNet, our method achieves a 0.1% improvement in the base session and an average improvement of 2.70% across all sessions. On CIFAR100, we achieve a 1.53% improvement in the base session and an average improvement of 3.09% across all sessions. On CUB200, we see a 0.18% improvement in the base session and an average improvement of 2.26% across all sessions. These results mean that the advantage of our method indeed gets larger during the incremental training. Moreover, as shown Table 3 of the manuscript and Tables 1 and 2 of the General Response, our method achieves superior performance in both base accuracy and incremental accuracy compared to NC-FSCIL. This indicates that the observed improvements are not solely due to the strong performance on base classes but also reflect better adaptability to incremental sessions. To provide a clearer view of stability across sessions, we also include the performance drop (PD) metric in Table 1,5 and 6 in the revised manuscript. For mini-ImageNet, the PD results show that our method outperforms other methods on ResNet-12 and is comparable to recent methods on ResNet-18. For CIFAR100, the PD results indicate that we achieve superior performance on ResNet-12 and remain comparable to other methods on ResNet-18 and ResNet-20. These findings demonstrate that our approach effectively enhances plasticity while maintaining competitive stability.
> > >
> > > ------
> > >
> > > [1] Neural Collapse Inspired Feature-Classifier Alignment for Few-Shot Class-Incremental Learning. ICLR2023.

---

> > > > ### Author Response · Authors · 2024-12-02
> > > > **To Reviewer 6ye5**
> > > >
> > > > Dear Reviewer  6ye5,
> > > >
> > > > Thank you for your careful review and thoughtful feedback! We have updated our submission based on your suggestions and provided detailed responses to the newly raised concerns. As the discussion phase is nearing its conclusion, could you please check our response to see if we address your concerns?  Please let us know if you have any other questions! We would be glad to answer them during the discussion period.
> > > >
> > > > Best regards,
> > > >
> > > > Authors

---

> > > > > ### Comment · Reviewer_6ye5 · 2024-12-02
> > > > >
> > > > > Dear Authors,
> > > > >
> > > > > Thank you for providing such detailed responses and a thorough revision.
> > > > > I think the revised manuscript explains the proposed method more effectively than the previous version, and the fairness issues in the experimental section have been fully addressed.
> > > > > As a result, I have decided to increase the rating to 'marginally below the acceptance threshold.'
> > > > > I still hesitate to give a positive rating because it seems that the experimental results do not sufficiently demonstrate the effectiveness of the proposed method.
> > > > > In particular, regarding 'Performance Decrease,' the proposed method is often outperformed by other approaches, indicating vulnerability to catastrophic forgetting.
> > > > > Since mitigating catastrophic forgetting is one of the primary goals of FSCIL, I do not believe the proposed method is effective in the FSCIL scenario.

---

> > > > > > ### Author Response · Authors · 2024-12-02
> > > > > > **Response to Reviewer 6ye5**
> > > > > >
> > > > > > Dear Reviewer  6ye5,
> > > > > >
> > > > > > Thank you for your constructive feedback and for raising the score of our submission. The PD metric reflects a model's ability to mitigate forgetting, but it represents only one aspect of overall performance. For instance, a model with consistently 0 accuracy across sessions would achieve a PD of 0, which might misleadingly appear optimal. Therefore, PD should always be considered alongside accuracy to provide a comprehensive assessment of an algorithm's effectiveness. We thank you again for helping improve our paper.
> > > > > >
> > > > > > Best regards,
> > > > > >
> > > > > > Authors

---

### Official Review · Reviewer_aBEC · 2024-10-28

**Soundness:** 3
**Presentation:** 4
**Contribution:** 3
**Rating:** 6
**Confidence:** 4

**Summary:**

This paper introduces a novel framework for few-shot class incremental learning (FSCIL) that addresses the challenges of plasticity and overfitting through meta-initialized adapters. The approach consists of three phases: meta-training adapters during the base session to obtain generalizable initial parameters, backbone pretraining with feature compactness loss to prevent feature space dispersion, and few-shot adaptation in incremental sessions where adapters are fine-tuned while preserving backbone knowledge. The framework demonstrates state-of-the-art performance across multiple benchmark datasets.

**Strengths:**

1. Novel architectural design that combines meta-learning with adapter modules in a way that enhances both plasticity and stability.

2. Three-phase training strategy that systematically addresses key FSCIL challenges. The strategy integrates meta-learning for initialization, feature space management during pretraining, and adaptation during incremental sessions.

3. Thorough empirical validation across multiple benchmark datasets with ablation studies. The experimental results demonstrate consistent performance improvements across different scenarios and provide insights into the contribution of each component.

**Weaknesses:**

1. While the paper presents reproducible results across three datasets, some existing techniques implemented in the program are not mentioned in the manuscript. For example, the rotation technique utilized in the implementation appears to draw from previous work [1]. The authors should include all the implementation details in the paper and illustrate how these designs facilitate their method.

[1] Learning with Fantasy: Semantic-Aware Virtual Contrastive Constraint for Few-Shot Class-Incremental Learning

2. A more detailed discussion of the novelty in the existing techniques combination should be made (meta-learning [2], SAM [3], and feature compactness loss in FSCIL [4][5][6]...). The paper should elaborate on how the specific combination and adaptation of these techniques contribute to improved performance and reduced forgetting in the FSCIL context, for example, highlighting any unique modifications or interactions between these designs.

[2] On First-Order Meta-Learning Algorithms;
[3] Sharpness-Aware Minimization for Efficiently Improving Generalization;
[4] Forward Compatible Few-Shot Class-Incremental Learning;
[5] Dycr: a dynamic clustering and recovering network for few-shot class-incremental learning;
[6] Few-Shot Class-Incremental Learning from an Open-Set Perspective;
...

3. The benchmark comparisons should include detailed network architecture specifications for all compared methods. In addition, the model parameter size should also be listed since additional parameters are introduced in MetaAdapter. This additional information would facilitate a better understanding of the relative complexity and computational requirements across different approaches.

**Questions:**

Please refer to the weaknesses of the paper.

**Details Of Ethics Concerns:**

Nil

---

> ### Author Response · Authors · 2024-11-22
> **Response to Reviewer aBEC**
>
> Thanks for your constructive comments! Below, we address your concerns point by point.
>
> ---
>
> **Q1**: While the paper presents reproducible results across three datasets, some existing techniques implemented in the program are not mentioned in the manuscript. For example, the rotation technique appears to draw from previous work [1].
>
> **A1**: Due to space constraints, we did not include the detailed augmentation strategies in the manuscript, but we will add them in the revised version. The augmentation strategies used in our work are consistent with those in prior studies.
>
> ---
>
> **Q2**: A more detailed discussion of the novelty in combining existing techniques is needed. The paper should elaborate on how these techniques contribute to improved performance and reduced forgetting, highlighting any unique modifications or interactions.
>
> **A2**: Our MetaAdapter framework introduces a novel combination of techniques specifically designed to balance plasticity and stability in few-shot class-incremental learning (FSCIL). To enhance plasticity, we leverage **Meta-Initialized Adapters**, which provide task-agnostic initialization for adapting to new tasks, and **Feature Compactness Loss (FCL)**, which compacts both inter-class and intra-class distances during the base session to reserve embedding space for future incremental tasks. To improve stability, we incorporate **Sharpness-Aware Minimization (SAM)** to locate flatter local minima during backbone training. This improves the generalization of the learned representation and reduces forgetting, particularly when adapter parameters are merged back into the backbone for final inference.
>
> ---
>
> **Q3**: The benchmark comparisons should include detailed network architecture specifications for all compared methods. Additionally, model parameter sizes should be listed, as MetaAdapter introduces additional parameters.
>
> **A3**: Detailed network architecture specifications for all compared methods have been included in Appendix  (Table 7). Additionally, we provide the parameter sizes for our method to illustrate its efficiency:
>
> - **ResNet-18**: 11.57M total parameters, 0.29M trainable parameters for adapters
> - **ResNet-12**: 12.81M total parameters, 0.32M trainable parameters for adapters
>
> The additional parameters introduced by MetaAdapter constitute less than 3% of the total parameter count.
>
> ---
>
> [1] Learning with Fantasy: Semantic-Aware Virtual Contrastive Constraint for Few-Shot Class-Incremental Learning. CVPR2023.
>
> [2] Few-Shot Class-Incremental Learning via Training-Free Prototype Calibration. NeurIPS2023.
>
> [3] Few-Shot Incremental Learning with Continually Evolved Classifiers. CVPR2021.

---

### Official Review · Reviewer_x4wA · 2024-10-31

**Soundness:** 2
**Presentation:** 3
**Contribution:** 2
**Rating:** 5
**Confidence:** 4

**Summary:**

This paper proposes to use meta-adapter to address the few-shot class incremental learning problem. Additionally, a feature compactness loss is introduced to help the accommodation of new categories. By leveraging the generalization capabilities of meta-learning training methods, the proposed method enhances the performance of few-shot class incremental tasks.

**Strengths:**

1. The paper is well-written and easy to follow.
2. The figures in the paper are concise and clear.
3. The idea of feature compactness loss and adapter integration is impressive.
4. The ablation studies in Table 2 and Figure 4 are extensive.

**Weaknesses:**

1. Two highly related works are not compared [1,2], which have the same task setting.
2. Although the feature compactness loss (FCL) is impressive and shows good performance improvement, the analysis of FCL is insufficient:
    - Why don't more similar inter-class feature representations affect classification?
    - Need further proof or experiment to demonstrate that more similar inter-class feature representations reserve embedding space. Additionally, will more similar inter-class feature representations possibly lead to the overall embedding space shrinking, similar to collapse in contrastive learning?
    - Need to prove that it is the reservation space that helps future tasks and not others. For example, reducing the number of novel categories in the few-shot adaptation task can also reserve space. Will this also improve performance?
    - The results using only FCL are missing in Table 2.
    - FCL is an interesting idea that I think needs further discussion.
3. It would be better to state how to expand $W_n^{t-1}$, Figure 2(b) only shows $W_n^{t}$.
4. typo in L95: "using this knowledge to improve leanring efficiency"


[1] Improved Continually Evolved Classifiers for Few-Shot Class-Incremental Learning. TCSVT2023. \
[2] Rethinking Few-shot Class-incremental Learning: Learning from Yourself. ECCV2024.

**Questions:**

1. Please address the Weaknesses.
2. Since the ViT backbones are receiving more attention, I wonder if meta-adapter and FCL could be migrated to methods with ViT backbone?

---

> ### Author Response · Authors · 2024-11-22
> **Response to Reviewer x4wA (1/2)**
>
> Thanks for your constructive comments! Below, we address your concerns point by point.
>
> ---
>
> **Q1**: Two highly related works [1, 2] with the same task setting are not compared.
>
> **A1**: We have updated our experimental results to include a detailed comparison with [1]  in our manuscript. However, we are unable to directly compare with [2] as it primarily utilizes a ViT-based architecture, which differs from the ResNet backbones used in our work. Nevertheless, our method demonstrates superior performance on both CIFAR100 and mini-ImageNet datasets.
>
> On mini-ImageNet
>
> |   Methods    |  0   |  1   |  2   |  3   |  4   |  5   |  6   |  7   |  8   |
> | :----------: | :--: | :--: | :--: | :--: | :--: | :--: | :--: | :--: | :--: |
> | Yourself [2] | 84.0 | 77.6 | 73.7 | 70.0 | 68.0 | 64.9 | 62.1 | 59.8 | 59.0 |
> | MetaAdapter  | 84.1 | 80.0 | 76.0 | 72.6 | 69.7 | 66.9 | 64.1 | 62.4 | 61.0 |
>
> On CIFAR100
>
> |   Methods    |  0   |  1   |  2   |  3   |  4   |  5   |  6   |  7   |  8   |
> | :----------: | :--: | :--: | :--: | :--: | :--: | :--: | :--: | :--: | :--: |
> | Yourself [2] | 82.9 | 76.3 | 72.9 | 67.8 | 65.2 | 62.0 | 60.7 | 58.8 | 56.6 |
> | MetaAdapter  | 84.1 | 78.9 | 75.2 | 71.6 | 68.3 | 65.3 | 63.5 | 61.5 | 59.2 |
>
> ---
>
> **Q2**: Why don't more similar inter-class feature representations affect classification?
>
> **A2:** Excessive feature compactness, caused by larger $w_{fcl}$ values, can negatively impact base session performance by reducing inter-class separation. By contrast, an appropriately  $w_{fcl}$ balances compactness and separation, preserving space for new-class adaptation while maintaining stability in the feature space.  In the **General Response**, we conducted experiments on both base and incremental accuracy across varying  $w_{fcl}$ values. From Table 1, base accuracy remains stable across different $w_{fcl}$ values, but excessively high values (e.g., $w_{fcl} = 3.0$) reduce performance due to over-compactness, which impacts base-class feature representation. From Table 2, incremental accuracy improves with moderate $w_{fcl}$ values (e.g., $w_{fcl}=1.0$), as this balances compactness and separation. Extremely high or low $w_{fcl}$ values reduce incremental accuracy due to excessive compactness or dispersion in the feature space.
>
> ---
>
> **Q3**: Need further proof or experiments to demonstrate that more similar inter-class feature representations reserve embedding space. Additionally, will more similar inter-class representations possibly lead to the overall embedding space shrinking, similar to collapse in contrastive learning?
>
> **A3**: To address this concern, we conducted additional experiments in the **General Response** to analyze the relationship between inter-class feature representations and embedding space reservation. Specifically, we conducted experiments on both base and incremental accuracy across varying  $w_{fcl}$ values (Table 1 and Table 2) and evaluated the **relative angular disparity** $T(f_\theta)$ between new-class samples and base-class prototypes (Table 3) and the **inter-class angular distance** among base-class prototypes (Table 4).  The results in Table 4 indicates that Inter-class angular distance among base-class prototypes decreases as $w_{fcl}$ increases, which shows that FCL effectively compacts the embedding space. As shown in Table 1 and Table 2, this compactness does not result in an overall collapse of the embedding space. Additionally, the $T(f_\theta)$ metric in Table 3 improves with moderate $w_{fcl}$, showing that reducing the angular separation among base-class features to an appropriate extent supports better adaptation to new classes.
>
> ------

---

> ### Author Response · Authors · 2024-11-22
> **Response to Reviewer x4wA (2/2)**
>
> **Q4**: Need to prove that it is the reservation of space that helps future tasks, not other factors.  For example, reducing the number of novel categories in the few-shot adaptation task can also reserve space. Will this also improve performance?
>
> **A4**: To address this, the ablation study has been updated to include results using only FCL in Table 2 of the manuscript, which demonstrate that the reservation of embedding space during the base session contributes to the observed performance improvements. And we would also like to clarify that this space reservation occurs mainly during the base session learning process and is irrelevant to the number of novel categories in subsequent tasks.
>
> ------
>
> **Q5**: The results using only FCL are missing in Table 2.
>
> **A5**: Thank you for catching this. We have updated Table 2 to include the results using only FCL. The updated results demonstrate that FCL alone improves the model’s performance.
>
> ------
>
> **Q6**: It would be better to state how to expand $W_{t-1}$, as Figure 2(b) only shows $W_t $.
>
> **A6**: Thank you for your suggestion. Figure 2(b) illustrates that $W_t$, the weight matrix for the current task $t$, consists of two components: (1) weights from all previous classes ($W^{t-1}$), which are retained as the learned representations from previous tasks, and (2) new weights for the current task ($w^t_1,w^t_2,w^t_3,\dots$),   which are derived from the feature means of the samples for each new class.
>
> ------
>
> **Q7**: Since the ViT backbones are receiving more attention, I wonder if MetaAdapter and FCL could be applied to methods with ViT backbones?
>
> **A7**: Your suggestion provides a valuable direction for future research. Our MetaAdapter framework is general and can be extended to Vision Transformer (ViT) backbones.  Specifically, adapting MetaAdapter would involve inserting the lightweight neural modules to work in parallel with self-attention layers. These modules can be implemented with residual connections and structured with a down-projection matrix, a nonlinear activation function, and an up-projection matrix to enable efficient representation learning. In addition, FCL is flexible and can be directly applied to ViT features, as it focuses on regulating feature dispersion. We believe that combining our MetaAdapter framework and FCL with ViT backbones has the potential to achieve superior performance on more complex tasks.
>
> ------
>
> [1] Improved Continually Evolved Classifiers for Few-Shot Class-Incremental Learning. TCSVT2023.
>
> [2] Rethinking Few-shot Class-incremental Learning: Learning from Yourself. ECCV2024.

---

> ### Comment · Reviewer_x4wA · 2024-11-25
>
> Thank you to the authors for their efforts in the rebuttal. While some of my concerns have been addressed, several issues still remain:
>   1. Regarding the $ W_{t-1} $ and $ W_t $ mentioned in Q6, I am actually curious that although their dimensions are different, both $ z^{t-1} $ and $ z^t $ have $ c^t $ weights in Eq.10. Is there perhaps some intermediate steps missing here?
>   2. According to the newly added results in Table 2 of the article, using only FCL yields better performance than SAM+FCL and is comparable to MIS+FCL. Could the authors please provide further explanation on this?
>   3. In Table 1 of the General Response, the results don't seem very stable. Together with Table 2, they show that under reasonable parameter values, FCL can enhance the model's generalization ability with minimal harm to its classification performance. This is a somewhat clever method, but I'm not quite sure about its stability and robustness. In addition, Could the authors provide the baseline performance in Tables 1 and 2 of the General Response, so that we can observe the range of $ w_{fcl} $ where FCL brings improvements?

---

> > ### Author Response · Authors · 2024-11-29
> > **Response to Reviewer x4wA**
> >
> > Thanks for your constructive comments! Below, we address your concerns point by point.
> >
> > ------
> >
> > **Q1**: Regarding the $W_{t−1}$ and $W_t$ mentioned in Q6,  Although their dimensions are different, both $\mathbf{z}^{t-1}$ and $\mathbf{z}^{t}$ have $c^{t}$ weights in Eq.10. Is there perhaps some intermediate steps missing here?
> >
> > **A1**: To clarify, during the $t$-th task, we only utilize $W_t$ for the classification weights. The term $\mathbf{z}^{t-1}$ refers to the logits produced solely by the backbone, as the adapter weights from the previous task have already been integrated into the backbone during the previous phase.
> >
> > ---
> >
> > **Q2**: According to the newly added results in Table 2 of the manuscript, using only FCL yields better performance than SAM+FCL and is comparable to MIS+FCL. Could you explain why this happens?
> >
> > **A2**: Thank you for raising this point. The results demonstrate that using FCL alone is indeed effective. When combined with SAM, the gradient-based perturbations, while beneficial for improving generalization, can slightly interfere with the compactness established by FCL. MIS can enhance the model's adaptability to few-shot tasks but this focus may lead to increased forgetting of previously learned tasks.  These factors explain why FCL alone yields better performance compared to SAM+FCL and remains comparable to MIS+FCL.
> >
> > ---
> >
> > **Q3**: The results in Table 1 show some fluctuations in FCL's performance, while Table 2 demonstrates that it enhances generalization on classification tasks under reasonable parameter values. However, concerns remain regarding the stability and robustness of the method. Baseline performance should be included in Tables 1 and 2 of the General Response to clearly illustrate the range of $w_{fcl}$ values where FCL demonstrates improvements.
> >
> > **A3**: Thank you for raising this point. We have updated the General Response to include the baseline performance of the NC-FSCIL [1], which has demonstrated strong performance in FSCIL. The results show that moderate $w_{fcl}$ values (e.g., around 1.0) consistently enhance both base and incremental accuracy compared to NC-FSCIL, which indicates the effectiveness of FCL in improving generalization and adaptability in FSCIL tasks.
> >
> > ---
> >
> > [1] Neural Collapse Inspired Feature-Classifier Alignment for Few-Shot Class-Incremental Learning. ICLR2023.

---

> > > ### Author Response · Authors · 2024-12-02
> > > **To Reviewer x4wA**
> > >
> > > Dear Reviewer  x4wA,
> > >
> > > Thank you for your careful review and thoughtful feedback! We have provided detailed responses to the newly raised concerns. As the discussion phase is nearing its conclusion, could you please check our response to see if we address your concerns?  Please let us know if you have any other questions! We would be glad to answer them during the discussion period.
> > >
> > > Best regards,
> > >
> > > Authors

---

> ### Comment · Reviewer_x4wA · 2024-12-03
> **Final response to authors**
>
> Thank you to the authors for their detailed responses. Most of my concerns have been addressed, and I have raised my score to 6: marginally above the acceptance threshold. However, considering the performance of the method in the ablation study and its potential to harm classification, I still believe that the stability and generalizability of the method could be improved.

---

> > ### Author Response · Authors · 2024-12-03
> > **Many thanks! We will improve our paper accordingly.**
> >
> > Dear Reviewer  x4wA,
> >
> > Thank you for your constructive feedback and for raising the score of our submission! We will continue to strive to improve our paper according to the constructive reviews. Many thanks for your time and consideration!
> >
> > Best regards,
> >
> > Authors

---

### Official Review · Reviewer_7Rcy · 2024-11-03

**Soundness:** 3
**Presentation:** 3
**Contribution:** 2
**Rating:** 6
**Confidence:** 3

**Summary:**

This paper introduces MetaAdapter, a framework designed to address challenges in Few-Shot Class Incremental Learning (FSCIL). By employing meta-learning, MetaAdapter initializes adapters that encode general knowledge, aiming to balance stability and plasticity during incremental learning. The training process involves three phases: meta-training adapters, applying a feature compactness loss to reserve space for future classes, and utilizing knowledge distillation during incremental sessions. The framework demonstrates state-of-the-art performance on benchmarks such as mini-ImageNet, CIFAR100, and CUB200.

**Strengths:**

1. **State-of-the-Art Performance:** MetaAdapter achieves competitive results on multiple FSCIL benchmarks, indicating its effectiveness in adapting to new classes with limited data.

2. **Comprehensive Framework:** The integration of meta-learning with feature compactness and knowledge distillation offers a holistic approach to incremental learning challenges.

3. **Well-Structured Presentation:** The paper is organized and clearly articulates the methodology, facilitating understanding.

**Weaknesses:**

1. **Limited Novelty in Meta-Learning Approach:** The application of meta-learning for adapter initialization resembles existing methods. Clarification on how this approach differs from established frameworks would strengthen the contribution.

2. **Training Complexity:** The three-phase training process, including feature compactness loss and sharpness-aware minimization, adds complexity. This may pose challenges for implementation in resource-constrained environments.

3. **Base Task Performance:** The framework appears to underperform in base classification tasks compared to other methods using the same backbone. Understanding the reasons for this discrepancy is crucial, as base task accuracy is vital for incremental learning stability.

4. **Typographical Errors:** Minor errors, such as “rataining” in Line 012 and “minImageNet” in Line 448, detract from the paper's professionalism. A thorough review to correct these is recommended.

5. **Limited Comparison with Recent Methods:** The paper compares MetaAdapter with only one method from 2024. Including a broader range of recent methods would provide a more comprehensive evaluation of its performance.

**Questions:**

Please refer to weakness

---

> ### Author Response · Authors · 2024-11-22
> **Response to Reviewer 7Rcy (1/2)**
>
> Thanks for your constructive comments! Below, we address your concerns point by point.
>
> ---
>
> **Q1**: Limited Novelty in Meta-Learning Approach. The application of meta-learning for adapter initialization resembles existing methods. Clarification on how this approach differs from established frameworks would strengthen the contribution.
>
> **A1:** Existing methods based on meta-learning, such as MetaFSCIL [1], focus on meta-testing simulation by sampling sequences of incremental tasks from base classes during the base phase. However, the inherent discrepancies in data distributions make it difficult to use base session data to accurately simulate real incremental sessions. Instead of mimicking the incremental evaluation process, we use meta-learning to obtain meta-initialized adapters that serve as a generalizable starting point for expanding and refining feature representations. This design helps reduce overfitting and enhances the model's plasticity for adapting to new tasks efficiently.
>
> ------
>
> **Q2**: Training Complexity. The three-phase training process, including feature compactness loss and sharpness-aware minimization, adds complexity. This may pose challenges for implementation in resource-constrained environments.
>
> **A2**: Our training process is divided into three phases:
>
> 1. **Phase 1 (Adapter Meta-Training)**: During the base session, we first construct few-shot tasks by randomly sampling instances from each base class and then train the adapters using meta-learning algorithms to obtain generalizable initial parameters.
>
> 2. **Phase 2 (Backbone Pretraining)**: In this phase, we focus on training the backbone. We introduce the feature compactness loss (FCL) to bring feature representations closer together and apply sharpness-aware minimization (SAM) to find flatter minima.
>
> 3. **Phase 3 (Few-Shot Adaptation)**: We only fine-tune the lightweight adapters for each new task to expand the current representations to encompass new class features.
>
> Phase 1 is computationally efficient because it involves only the limited size of few-shot tasks. And the feature compactness loss and sharpness-aware minimization are applied solely during Phase 2 to train the backbone. In Phase 3, we fine-tune the adapters with just 1-5 iterations, which enables rapid adaptation for few-shot new classes.
>
> ---
>
> **Q3**: Base Task Performance. The framework appears to underperform in base classification tasks compared to other methods using the same backbone. Understanding the reasons for this discrepancy is crucial, as base task accuracy is vital for incremental learning stability.
>
> **A3:**  We would like to clarify that our framework achieves comparable performance to NC-FSCIL [2] and outperforms other compared methods on base classification tasks. In the context of FSCIL, while base task performance is important, the most critical challenge is mitigating catastrophic forgetting. Simply optimizing for base session performance may not lead to better results in the final session, as overfitting to base classes can reduce adaptability to new tasks.
>
> Furthermore, the improved base task performance can be attributed to several factors. We utilize a prototype-based classifier with cosine similarity, where we re-scale the normalized features using a preset scale factor $\tau$. The choice of $\tau$ controls the separation between classes, which improves feature discrimination. For CIFAR100 and miniImageNet, we set $\tau = 64$, while for CUB200, we use $\tau = 32$ to better adapt to the characteristics of each dataset. Furthermore, we incorporate sharpness-aware minimization (SAM) to find flatter local minima by adding gradient-based perturbations to the parameters, which enhances the model’s generalization in base classification tasks.
>
> ------

---

> > ### Author Response · Authors · 2024-11-22
> > **Response to Reviewer 7Rcy (2/2)**
> >
> > **Q4**: Limited Comparison with Recent Methods. The paper compares MetaAdapter with only one method from 2024. Including a broader range of recent methods would provide a more comprehensive evaluation of its performance.
> >
> > **A4**: We incorporated CEC+ [3] and OrCo [4] as baselines across all three datasets (mini-ImageNet, CIFAR100, and CUB200), while comparisons with C-FSCIL [5] were conducted on mini-ImageNet and CIFAR100, as the original work did not evaluate on CUB200. These updated results have been included in the experimental results of the manuscript. As can be seen from the tables,  MetaAdapter performs better than other methods across all three datasets.
> >
> > On mini-ImageNet
> >
> > |   Methods   |   0   |   1   |   2   |   3   |   4   |   5   |   6   |   7   |   8   |
> > | :---------: | :---: | :---: | :---: | :---: | :---: | :---: | :---: | :---: | :---: |
> > |    CEC+     | 82.65 | 77.82 | 73.59 | 70.24 | 67.74 | 64.82 | 61.91 | 59.96 | 58.35 |
> > |   C-FSCIL   | 76.40 | 71.14 | 66.46 | 63.29 | 60.42 | 57.46 | 54.78 | 53.11 | 51.41 |
> > |    OrCo     | 83.30 | 70.80 | 66.90 | 64.32 | 62.28 | 60.46 | 58.40 | 58.02 | 58.08 |
> > | MetaAdapter | 84.12 | 79.95 | 75.97 | 72.61 | 69.68 | 66.88 | 64.12 | 62.39 | 61.01 |
> >
> > On CIFAR100
> >
> > |   Methods   |   0   |   1   |   2   |   3   |   4   |   5   |   6   |   7   |   8   |
> > | :---------: | :---: | :---: | :---: | :---: | :---: | :---: | :---: | :---: | :---: |
> > |    CEC+     | 81.25 | 77.23 | 73.30 | 69.41 | 66.69 | 63.93 | 62.16 | 59.62 | 57.41 |
> > |   C-FSCIL   | 77.47 | 72.40 | 67.47 | 63.25 | 59.84 | 56.95 | 54.42 | 52.47 | 50.47 |
> > |    OrCo     | 80.08 | 71.46 | 64.95 | 58.65 | 57.60 | 56.68 | 56.16 | 54.62 | 52.19 |
> > | MetaAdapter | 84.05 | 78.86 | 75.16 | 71.64 | 68.29 | 65.31 | 63.54 | 61.52 | 59.20 |
> >
> > On CUB200
> >
> > |   Methods   |   0   |   1   |   2   |   3   |   4   |   5   |   6   |   7   |   8   |   9   |  10   |
> > | :---------: | :---: | :---: | :---: | :---: | :---: | :---: | :---: | :---: | :---: | :---: | :---: |
> > |    CEC+     | 79.46 | 76.11 | 73.12 | 69.31 | 67.97 | 65.86 | 64.50 | 63.83 | 62.20 | 62.00 | 60.97 |
> > |    OrCo     | 75.59 | 72.74 | 64.58 | 60.12 | 60.16 | 58.04 | 58.41 | 57.96 | 56.97 | 57.99 | 57.93 |
> > | MetaAdapter | 80.63 | 76.85 | 73.62 | 69.75 | 69.13 | 66.23 | 65.67 | 64.51 | 62.29 | 62.58 | 61.70 |
> >
> > ------
> >
> > [1] MetaFSCIL: A Meta-Learning Approach for Few-Shot Class Incremental Learning. CVPR2022.
> >
> > [2] Neural Collapse Inspired Feature-Classifier Alignment for Few-Shot Class-Incremental Learning. ICLR2023.
> >
> > [3] Improved Continually Evolved Classifiers for Few-Shot Class-Incremental Learning. TCSVT2023.
> >
> > [4] Hersche et al., Constrained few-shot class-incremental learning, CVPR2022.
> >
> > [5] OrCo: Towards Better Generalization via Orthogonality and Contrast for Few-Shot Class-Incremental Learning. CVPR2024.

---

> > > ### Comment · Reviewer_7Rcy · 2024-11-24
> > > **Post-Rebuttal Review**
> > >
> > > Most of my concerns have been adequately addressed in the rebuttal. I am willing to raise my recommendation to a 6: marginally above the acceptance threshold.

---

> > > > ### Author Response · Authors · 2024-11-24
> > > > **Many thanks! We will improve our paper accordingly.**
> > > >
> > > > Dear Reviewer 7Rcy,
> > > >
> > > > Thank you very much for your kind reply! We will continue to strive to improve our paper according to the constructive reviews.
> > > >
> > > > Best regards,
> > > >
> > > > Authors

---

### Author Response · Authors · 2024-11-22
**General Response To All Reviewers (1/2)**

We would like to sincerely thank all the reviewers for their valuable feedback and constructive suggestions. Below, we provide an explanation of our feature compactness loss (FCL) and new experimental results to address your concerns.

------

**Q**: While the feature compactness loss (FCL) demonstrates notable performance improvements, its motivation and comparisons with existing methods need further clarification and exploration.

**A**: To clarify, we first summarize the core strategies of existing methods:

1. **FACT [1]** FACT utilizes manifold mixup during the base session to create virtual new classes, thereby making room for new categories in subsequent incremental learning stages.
2. **ALICE [2]**: This method employs an angular penalty to simultaneously minimize the distance between intra-class feature vectors while maximizing the distance between inter-class vectors.
3. **SAVC [3]**: SAVC leverages the MoCo [4] framework for self-supervised contrastive learning  to enhance feature representation, with the goal of increasing inter-class distance and reduce intra-class distance during the base session.

As CLOM [5] suggests, while increasing inter-class distances may improve base-class performance,  it often leads to overfitting on base classes, which can harm generalization to few-shot new classes.  Additionally, this strategy overlooks the issue of **Minority Collapse [6]**, where few-shot class features cluster too tightly in imbalanced scenarios, potentially resulting in performance degradation for new classes. Our feature compactness loss (FCL) takes a different approach by compacting both inter-class and intra-class distances during the base session. This design prevents the embedding space from becoming overly dispersed, effectively enabling the model to better adapt to new tasks in future incremental sessions.

To substantiate the effectiveness of our feature compactness loss (FCL), we have conducted additional experiments as follows:

- Base accuracy for different datasets across varying $w_{fcl}$ values.
- Incremental accuracy for different datasets across varying $w_{fcl}$ values.
- Relative angular disparity $T(f_{\theta})$ [7] for different datasets across varying $w_{fcl}$ values. This metric measures the relative angular disparity between new-class samples and base-class prototypes to assess adaptability for new categories.
- Inter-class angular distance among base-class prototypes for different datasets across varying $w_{fcl}$ values.

---

> ### Author Response · Authors · 2024-11-22
> **General Response To All Reviewers (2/2)**
>
> **Table 1: Final Base Accuracy Across Datasets for Varying $w_{fcl}$ Values ($w_{kd} = 1.0$)**
>
> |Dataset|$w_{fcl}$|0.25|0.5|0.75|1.0|1.5|2.0|2.5|3.0|NC-FSCIL [8]|
> |:-----------:|:-------:|:---:|:---:|:---:|:---:|:---:|:---:|:---:|:---:|:----------:|
> |mini-ImageNet|Base|79.02|79.37|79.68|79.68|78.30|78.90|78.57|77.88|75.77|
> |CIFAR100|Base|76.41|76.90|76.86|75.17|73.02|73.86|73.78|73.22|73.98|
> |CUB200|Base|77.79|77.57|77.36|76.86|75.85|75.94|76.37|75.84|76.19|
>
> Table 1 shows that base accuracy is relatively stable across varying $w_{fcl}$ values for all datasets. However, excessively high values (e.g., $w_{fcl}= 3.0$) lead to a slight drop in performance due to over-compactness in the embedding space, which can harm the representation of base classes.
>
> **Table 2: Final Incremental Accuracy Across Datasets for Varying $w_{fcl}$ Values ($w_{kd} = 1.0$)**
>
> |Dataset|$w_{fcl}$|0.25|0.5|0.75|1.0|1.5|2.0|2.5|3.0|NC-FSCIL [8]|
> |:-----------:|:---------:|:---:|:---:|:---:|:---:|:---:|:---:|:---:|:---:|:----------:|
> |mini-ImageNet|Incremental|29.60|30.25|32.22|33.00|33.33|30.17|29.80|29.38|31.77|
> |CIFAR100|Incremental|31.05|31.25|31.53|35.25|36.07|35.57|35.25|31.15|29.30|
> |CUB200|Incremental|45.39|46.09|47.09|47.11|47.22|47.45|46.89|46.83|43.07|
>
> As presented in Table 2, incremental accuracy benefits from moderate $w_{fcl}$ values (e.g., $w_{fcl}=1.0$), as this balances compactness and separation.  Extremely high or low $w_{fcl}$ values reduce performance due to excessive compactness or dispersion in the feature space.
>
> As presented in Table 2, incremental accuracy benefits from moderate $w_{fcl}$ values (e.g., $w_{fcl}=1.0$), as this balances compactness and separation.  Extremely high or low $w_{fcl}$ values reduce performance due to excessive compactness or dispersion in the feature space.
>
> **Table 3: Relative Angular Disparity $T(f_\theta)$ for New-Class Adaptation Across Datasets for Varying $w_{fcl} $ Values ($w_{kd} = 1.0 $)**
>
> |$w_{fcl}$|mini-ImageNet|CIFAR100|CUB200|
> |:--------:|:-----------:|:------:|:----:|
> |0.25|0.8212|0.7649|0.7243|
> |0.5|0.8388 |0.7492|0.7420|
> |0.75|0.8635|0.7579|0.7645|
> |1.0|0.9015|0.7791|0.7931|
> |1.5|0.9353|0.8229|0.8124|
> |2.0|0.8267|0.6631|0.7420|
> |2.5|0.8234|0.6626|0.7136|
> |3.0|0.7940|0.5968|0.7016|
>
> As presented in Table 3, the $T(f_\theta)$ metric improves with moderate $w_{fcl}$, indicating that reducing the angular separation among base-class features  appropriately supports better adaptation to new classes. However, excessively high $w_{fcl}$ compresses base-class features too much and reduces the model's ability to adapt to new-class representations effectively.
>
> **Table 4: Inter-class Angular Distance Among Base-Class Prototypes Across Datasets for Varying $w_{fcl} $ Values ($w_{kd} = 1.0 $)**
>
>
> | $w_{fcl} $ | mini-ImageNet | CIFAR100 | CUB200 |
> | :--------: | :-----------: | :------: | :----: |
> |0.25|0.4252|0.4613|0.6375|
> |0.5|0.3952|0.4568|0.5561|
> |0.75|0.3826|0.4389|0.5255|
> |1.0|0.3588|0.4367|0.4983|
> |1.5|0.3238|0.3944|0.4730|
> |2.0|0.3255|0.3254|0.4496|
> |2.5|0.3323|0.3353|0.4404|
> |3.0|0.3055|0.3132|0.4304|
>
> From Table 4, Inter-class angular distance among base-class prototypes decreases as $w_{fcl}$ increases, which shows that FCL effectively compacts the embedding space.
>
>
>
> We hope that our response could relieve your concerns. Please let us know if you have any further questions or suggestions.
>
>
>
> [1] Forward compatible few-shot class-incremental learning. CVPR2022.
>
> [2] Few-Shot Class-Incremental Learning from an Open-Set Perspective.  ECCV2022.
>
> [3] Learning with Fantasy: Semantic-Aware Virtual Contrastive Constraint for Few-Shot Class-Incremental Learning. CVPR2023.
>
> [4] Momentum contrast for unsupervised visual representation learning. CVPR2020.
>
> [5] Margin-based few-shot classincremental learning with class-level overfitting mitigation. NeurIPS2022.
>
> [6] Neural collapse inspired attraction--repulsion-balanced loss for imbalanced learning. Neurocomputing2023.
>
> [7] CLOSER: Towards Better Representation Learning for Few-Shot Class-Incremental Learning. ECCV2024.
>
> [8] Neural Collapse Inspired Feature-Classifier Alignment for Few-Shot Class-Incremental Learning. ICLR2023.
>
> ------

---

### Meta-Review · Area_Chair_5RFH · 2024-12-18

**Metareview:**

This submission received two negative sores and two positive scores after rebuttal. After carefully reading the paper, the review comments, the AC can not recommend the acceptance of this submission, as the average score is under the threshold bar and the concerns about the proposed approach remain. The AC also recognizes the contributions confirmed by the reviewers, and encourages the authors to update the paper according to the discussion and submit it to the upcoming conference.

**Additional Comments On Reviewer Discussion:**

This submission was fully discussed during the rebuttal period. While most concerns of reviewer#7Rcy and aBEC were solved, those (novelty, motivation and effectiveness) from Reviewer #ma3t, #6ye5 and #x4wA were only partially solved.

---

### Decision · Program_Chairs · 2025-01-22

Reject